**A Stalagmite Test of North Atlantic SST and Iberian Hydroclimate Linkages over the Last Two Glacial Cycles**

Rhawn F. Denniston[a*], Amanda N. Houts[a1], Yemane Asmerom[b], Alan D. Wanamaker, Jr.[c], Jonathon A. Haws[d], Victor J. Polyak[b], Diana L. Thatcher[c], Setsen Altan-Ochir[a], Alyssa C. Borowske[a2], Sebastian F.M. Breitenbach[e], Caroline C. Ummenhofer[f], Frederico T. Regala[g], Michael M. Benedetti[h], Nuno Bicho[i]

[a] Department of Geology, Cornell College, Mount Vernon, Iowa 52314 USA
[b] Department of Earth and Planetary Sciences, University of New Mexico, Albuquerque, New Mexico 87131 USA
[c] Department of Geological and Atmospheric Sciences, Iowa State University, Ames, Iowa 50011 USA
[d] Department of Anthropology, Louisville University, Louisville, Kentucky 40208 USA
[e] Institute for Geology, Mineralogy, and Geophysics, Ruhr-University Bochum 44801 Germany
[f] Department of Physical Oceanography, Woods Hole Oceanographic Institution, Woods Hole, Massachusetts 02543 USA
[g] Associação de Estudos Subterrâneos e Defesa do Ambiente, Torres Vedras, Portugal
[h] Department of Geography and Geology, University of North Carolina Wilmington, Wilmington, North Carolina 28403 USA
[i] Center for Archaeology and Evolution of Human Behaviour, Universidade do Algarve, Faro, Portugal

[*] corresponding author

[1] current address: Department of Earth Sciences, University of New Hampshire, Durham, New Hampshire 03824 USA

[2] current address: Department of Ecology and Evolutionary Biology, University of Connecticut, Storrs, Connecticut 06269 USA

**Keywords**

Iberia, hydroclimate, stalagmite, oxygen isotope, carbon isotope, $\delta^{234}U$, pollen, sea surface temperature

**Abstract**

Close coupling of Iberian hydroclimate and North Atlantic sea surface temperature (SST) during recent glacial periods has been identified through the analysis of marine sediment and pollen grains co-deposited on the Portuguese continental margin. While offering precisely correlatable records, these time series have lacked a directly-dated, site-specific record of continental Iberian climate spanning multiple glacial cycles as a point of comparison. Here we present a high-resolution, multi-proxy (growth dynamics and $\delta^{13}$C, $\delta^{18}$O, and $\delta^{234}$U values) composite stalagmite record of hydroclimate from two caves in western Portugal across the majority of the last two glacial cycles (~220 ka). At orbital and millennial scales, stalagmite-based proxies for hydroclimate proxies covaried with SST, with elevated $\delta^{13}$C, $\delta^{18}$O, and $\delta^{234}$U values and/or growth hiatuses indicating reduced effective moisture coincident with periods of lowered SST during major ice-rafted debris events, in agreement with changes in palynological reconstructions of continental climate. While in many cases the Portuguese stalagmite record can be scaled to SST, in some intervals the magnitudes of stalagmite isotopic shifts, and possibly hydroclimate, appear to have been somewhat decoupled from SST.

**1. Introduction**

The Portuguese continental margin is an important location for understanding variations in paleoceanographic conditions over orbital and millennial-scales (Hodell et al., 2013; Voelker and de Abreu, 2011). Here, marine sediments record basin-wide oceanographic signals while co-deposited pollen grains track coeval vegetation changes occurring across Iberia. Integrated analysis of these proxies has revealed a close coupling of North Atlantic SST, regional climate, and Iberian ecosystems during the last three glacial cycles, including changes in vegetation dynamics (Sánchez Goñi et al., 2002; Tzedakis et al., 2004; Roucoux et al., 2006; Martrat et al., 2007; Naugthon et al., 2007; Sánchez Goñi et al., 2008), atmospheric circulation (Sánchez Goñi et al., 2013), and fire frequency (Daniau et al., 2007). One commonly applied palynological metric is the abundance of temperate tree pollen, which rises during warm and wet conditions associated both with interglacials and Greenland interstadials, concomitant with shifts in Iberian margin SST (Sánchez Goñi et al., 2002; Tzedakis et al., 2004; Combourieu-Nebout et al., 2009; Fletcher et al., 2010; Chabaud et al., 2014). However, the nature of such land-sea connections is partially obscured by the size of catchments from which the pollen are derived, with some

reaching into central Iberia and spanning a range of environmental settings subject to varying
climatic influences (Martin-Vide and Lopez-Bustins, 2006; Naughton et al., 2007) (Fig. 1).

Testing the links between terrestrial and marine systems benefits from continental climate

archives that provide precisely-dated and high resolution rainfall-sensitive time series spanning
tens of millennia, but such records remain rare in Iberia, particularly near the west Iberian
margin (Fletcher et al., 2010; Moreno et al., 2012; Stoll et al., 2013). Here we present a
composite stalagmite record of four proxies for hydroclimate – growth dynamics and $\delta^{13}$C, $\delta^{18}$O,
and $\delta^{234}$U values – spanning the majority of the last and penultimate glacial cycles (~220 ka) at
two cave sites in western Portugal. These time series offer a rare, site-specific continental record
capable of examining the coherence of SST controls on Iberian climate and ecosystem dynamics
across glacial and interglacial periods. The new record provides a continental perspective of
hydroclimate dynamics linked to regional oceanographic conditions.

**2. Samples and Regional Setting**
*2.1 Environmental Setting*

We report the analysis of five stalagmites (BG41, BG66, BG67, BG611, BG6LR) from

Buraca Gloriosa (BG; 39º32'N, 08º47'W; 420 m a.s.l.) and one stalagmite (GCL6) from Gruta
do Casal da Lebre (GCL; 39˚18'N, 9˚16'W; 130 m a.s.l.), two caves in western Portugal (Fig. 1).
Environmental conditions in BG and GCL are well suited for speleothem paleoclimate
reconstruction (see below). BG and GCL are located within the Meso-Mediterranean bioclimatic
zone that dominates much of Iberia (Fig. 1). This region is characterized by strong seasonality,
with warm, dry summers and cool, wet winters (Fig. 2) associated with the winter westerlies
(Blanco Castro et al., 1997). In contrast, the Atlantic zone, north of the Douro River, is cooler,
wetter, and less strongly seasonal. In the Pleistocene, the transition between these zones likely
shifted southward with Mediterranean-type vegetation restricted to refugia (Rey Benayas and
Scheiner, 2002).

Over interannual scales, the hydroclimate of Iberia is tightly coupled with the winter

North Atlantic Oscillation (NAO) (Fig. 3), an atmospheric dipole that strongly influences
precipitation across much of western Europe and that more broadly reflects the strength and
positioning of the Azores high pressure system, which steers storm tracks contained within the
westerlies into or north of Iberia (e.g., Trigo et al, 2002; Paredes et al., 2006; Trouet et al., 2009;
Cortesi et al., 2014). The NAO is typically measured as the NAO index, which is calculated
using atmospheric pressure differences between Iceland and Lisbon (or the Azores) (Barnston
and Livezey, 1987). The nature of the influence of the NAO varies across Iberia, but it is
strongly correlated to rainfall in western Portugal (Fig. 3), with a positive NAO index associated
with a steeper pressure gradient and elevated Iberian aridity. Iberian precipitation has also been
linked to SST in regions ranging from the western North Atlantic to the Iberian margin (Lorenzo
et al., 2010) where ocean circulation is dominated by the south-flowing Portugal Current and the
near-coastal, north-flowing Iberian Poleward Current, two systems that transport pollen from
river mouths along the continental shelf (Fig. 1).

*2.2 Cave Settings*
Buraca Gloriosa cave is located near the town of Alvados, 30 km from the Atlantic
Ocean, within middle Jurassic limestones of the Estremadura Limestone Massif (Rodrigues and
Fonseca, 2010), a topographically distinct region in central Portugal (Fig. 1). The ~35 m-long
cave is accessed through a single, small (~0.5 m$^2$) entrance at the top of a collapse at the base of
a 30 m-high escarpment (Fig. 4). The cave is well decorated although little active growth is
occurring today. Vegetation above the cave is primarily shrubs, small trees, and mosses, hosted
by a thin (0-10 cm) and highly organic soil layer.
Gruta do Casal da Lebre overlooks the coastal town of Peniche and is hosted by upper
Jurassic limestones. The cave is 130 m long and contains a single, one m$^2$ entrance that opens
onto a 7 m vertical shaft (Fig. 4). This entrance has been closed with a solid metal door in recent
decades in order limit access to the cave, and this modification likely has reduced air exchange in
GCL relative to its original state. Like BG, GCL hosts little active calcite deposition, but
contains numerous fossil stalagmites and stalactites. The vegetation over the cave has been
replaced in recent decades by stands of eucalyptus that grow in thin (<1-5 cm), clay-rich soils.

*2.3 Pollen Sources*
Pollen deposited on the west Iberian margin is sourced primarily from vegetation
inhabiting the watersheds of the major west-flowing stream systems draining Portugal and Spain,
which are (from north to south) the Douro, Tagus, and Sado rivers. The areas encompassed by
these streams are large (79,000, 81,000, and 7,650 km$^2$, respectively) and span a variety of
elevations. The Tagus and Sado are primarily responsible for pollen deposited southwest of
Portugal, while the Douro plays an important role in delivering pollen to the more northwesterly
sites (Fig. 1). Prevailing wind patterns likely prevent substantial transport of pollen from Iberia
to the western Portuguese margin (Naughton et al., 2007). The pollen data presented here were
collected in three closely spaced cores from the southwest Iberian margin: MD01-2443: 250-194
ka (Roucoux et al, 2006; Tzedakis et al., 2004); MD01-2444: 193-136 ka (Margari et al., 2010;
Margari et al., 2014); MD95-2042: 141-1 ka (Sánchez Goñi et al., 2008; Sánchez Goñi et al.,
2013) (Fig. 1) and are integrated here into a single time series.

**3. Materials and Methods**
*3.1 Environmental Monitoring*
Environmental conditions were measured at both cave sites over a multi-year period, with
data recorded in two-hour intervals near the areas where the stalagmites were deposited.
Temperature and relative humidity were obtained using HOBO U23 automated sensors while
barometric pressure was recorded with HOBO U20L loggers. Drip rates were monitored at BG
with Stalagmate acoustic drip counters (Collister and Mattey, 2008).

*3.2 Uranium-Series Dating*
Stalagmite chronologies were constructed with a total of 69 $^{230}$Th dates obtained at the
University of New Mexico (Table 1) using the methods of Asmerom et al. (2010). For dating of
stalagmite carbonate, powders ranging from 100-200 mg were weighed, dissolved in 15N nitric
acid, spiked with a mixed $^{229}$Th-$^{233}$U-$^{236}$U tracer, and processed using column chemistry
methods. U and Th fractions were dissolved in 5 ml of 3% nitric acid and transferred to analysis
tubes for measurement on a Thermo Neptune MC-ICP-MS. U and Th solutions were aspirated
into the Neptune using a Cetac Aridus II low flow desolvating nebulizer and run as static
routines. All isotopes of interest were measured in Faraday cups, except for $^{234}$U and $^{230}$Th,
which were measured in the secondary electron multiplier (SEM). Gains between the SEM and
the Faraday cups were determined using standard solutions of NBL-112 for U and an in-house
$^{230}$Th-$^{229}$Th standard for Th that was measured after every fifth sample; chemistry blanks reveal
U and Th blanks below 20 pg. Ages are reported using two standard deviation errors.
For BG stalagmites, corrections were made for unsupported $^{230}$Th using a $^{230}$Th/$^{232}$Th
ratio of 13.5 ppm (±50%), a value determined from isotopic analysis of cave dripwater. To
obtain this value, 108 ml of dripwater were transferred into six 30 ml Teflon beakers. These
beakers were fluxed in 6N HCl for an hour, rinsed, and heated gently on a hotplate until
approximately 1-2 ml of fluid remained in each. All solutions were then combined into a single
30 ml Teflon beaker, spiked with the same tracer described above (which contains HF), fluxed,
and then taken to complete dryness. The resulting precipitate was dissolved with 15N HNO$_3$,
dried down, dissolved again in 7N HNO$_3$, and processed with the same column chemistry
methods used for the stalagmite samples. We lack independent constraints on the initial Th ratio
for the GCL stalagmite, and thus apply the default value of 4.4 ppm (±50%). This difference in
the initial Th ratio impacts the corrected ages of GCL6 by 0.5-3.0 kyr relative to the value used
for BG, and thus does not meaningfully influence our interpretations.
Age models were developed via multiple polynomial interpolations between dated
intervals using the COPRA age modeling software (Breitenbach et al., 2012) (Fig. 5). Aside from
providing age models, COPRA also yields mean modeled stable isotope values and confidence
intervals (Supp. Fig. S1). Here we rely primarily here on the original $\delta^{18}$O and $\delta^{13}$C values
because COPRA-derived median values reflect statistically robust variations, but reduce to some
degree the range of isotopic variability. For COPRA, a dummy age was included in the age
model for BG41 in order to extrapolate below the hiatus, which is only possible with at least two
dated points. The value of this dummy age was based on the assumption that it maintains a
stratigraphically correct slope (i.e. higher sections of the stalagmite represent younger material).
The dummy age was applied a conservative error, meaning that it was as large as possible
without causing stratigraphic inversion with respect to the bounding ages.

*3.3 Stable Isotope Ratios*
A total of 1,510 stable isotope analyses were performed on calcite samples milled from
the central axis of each stalagmite. After milling, powders were weighed (~200 μg) and
transferred to reaction vessels that were flushed with ultra-pure helium. Samples were then
digested using >100% H$_3$PO$_4$ and equilibrated overnight (~16 hours) at 34˚C before being
analyzed. Isotopic ratios were measured using a GasBench II with a CombiPal autosampler
coupled to a Thermo Finnigan Delta Plus XL mass spectrometer at Iowa State University. A
combination of internal and external standards was run after every fifth sample, as well as before
and after each batch, in order to ensure reproducibility. Oxygen and carbon isotope ratios are
presented in parts per mil (‰) relative to the Vienna Pee Dee Belemnite carbonate standard
(VPDB). Average precision for both $\delta^{13}C$ and $\delta^{18}O$ analyses is better than ±0.1‰ (1$\sigma$).

For isotopic analyses of soil organic matter and vegetation collected from above the

caves, samples were dried, crushed, and transferred to tin boats. Carbon isotopic ratios were
measured using a Thermo Finnegan Delta Plus XL mass spectrometer in continuous flow mode
coupled with a Costech Elemental Analyzer. Caffeine (IAEA-600), cellulose (IAEA-CH-3), and
acetanilide (laboratory standard) isotopic standards yielded an average analytical uncertainty for
carbon of ±0.09‰ 1$\sigma$ (VPDB). Dripwater samples were measured using a Picarro L2130-i
Isotopic Liquid Water Analyzer, with autosampler and ChemCorrect software. Each sample was
measured six times, with only the last three injections used to determine isotopic values in order
to minimize memory effects. Three reference standards (VSMOW, IAEA-OH-2, IAEA-OH-3)
were used for regression-based isotopic corrections and to assign the data to the appropriate
isotopic scale. Reference standards were measured at least once every five samples. The average
analytical uncertainty for $\delta^{18}O$ measurements was ± 0.1‰ 1$\sigma$ (VSMOW).

*3.4 Stalagmite Mineralogy and Fabrics*

The calcite comprising the BG samples ranges across a variety of fabrics including a

faster-growing, white, fibrous form and a slower-growing, dense, clear structure (Fig. 6; Supp.
Fig. S2). In some samples, sharp changes between the two forms within the same growth
horizons mark intervals of recrystallization during which U/Th ages are highly inconsistent, and
these intervals were excluded from our data set. BG6LR, which grew discontinuously over much
of the last glacial cycle, suffered from alteration of early and middle Holocene material, which
was therefore excluded from this analysis. BG67 is characterized primarily by fibrous calcite that
has been recrystallized to clear, dense calcite in a narrow band descending through its core. U/Th
dates from the fibrous calcite on the margins of the growth surface reveal open system behavior
and thus this portion of BG67 was excluded. Recrystallization is evident in portions of GCL6
(particularly just above its base) and BG66 but the consistency of U/Th dates and the trends in
stable isotopes suggest that this alteration may have occurred soon after original deposition. We
tested whether these altered sections retain reliable paleoclimatic information by analyzing stable
isotopes along partial transects located just outside the zones of recrystallization (Fig. 6).
Because stable isotopic values and trends between these transects were consistent (within the
analytical errors), we retained these sections in the time series. Growth position changed at
numerous times in several of these stalagmites, and our sampling strategy accounted for these
changes so as to consistently collect samples for stable isotopic analysis from the top surface
(cap) of each stalagmite rather than the margins.

**4. Results**
*4.1 Environmental Monitoring*
Temperature and relative humidity collected inside both caves document environmental
conditions over a multi-year period. Relative humidity remained largely stable at ~100% in both
caves. Temperatures, while different at the two sites, exhibited similar seasonal variability that
approximates the mean average temperature of the region (14.2±0.4°C at BG and 16.2±0.3°C at
GCL for August 2012-January 2018) (Fig. 7).
Dripwater was collected at BG both over the course of minutes during site visits on four
separate occasions (November 2014, October 2015, March 2016, January 2018) and as months-
long integrated samples. A total of 25 dripwater samples were analyzed for stable isotopic
values. Dripwater $\delta^{18}O$ values range from -2.4‰ to -4.6‰, with a mean of -3.8±0.8‰ (Supp
Table 1), although as the timing of site visits varied, this value clearly is impacted by seasonal
controls on precipitation (and thus infiltration) oxygen isotope values. Drip rates were measured
for much of the period spanning June 2014 to January 2018 (for a total of ~36 months) and
exhibit seasonal variations tied to the winter wet and summer dry seasons, as well as individual
rain events (Fig. 7).

*4.2 U-Th Dates and Age Models*
$^{234}U$-$^{230}Th$ dating of BG and GCL stalagmites reveals growth across approximately three
quarters of the last 220 ka, with periods of deposition interrupted by numerous hiatuses of
varying length, with the longest gaps from 160-147, 97-87, 72-60, 41-36, 32-30, and 17-15 ka
(Fig. 5 and 6; Supp. Fig. S3). These features, coupled with repeated changes in growth direction
and high $^{232}Th$ abundances in select sections, complicate construction of a chronology in some
intervals. Macroscopic petrographic discontinuities suggest the presence of several short-lived
hiatuses, but these were included as gaps in the age models only where U/Th dates reveal an
identifiable temporal offset. For example, the marine isotope stage (MIS) 6/5e boundary
recorded by stalagmite BG67 is marked by both a change in drip position and a sharp transition
from dense, clear calcite to a white, fibrous form. Taken together, it is clear that a hiatus of some
duration occurred at this time. However, these isotope data are presented as being uninterrupted
given the continuity of $\delta^{18}O$ values and no U/Th evidence for a long-lived hiatus (Fig. 6).

*4.3 Assessing Equilibrium in Speleothem $\delta^{18}O$ and $\delta^{13}C$ Values*

We used two approaches to assess the fidelity of BG/GCL carbon and oxygen isotopes as

records of past environmental variability. First, Hendy Tests, in which stalagmite isotopic ratios
must satisfy two criteria in order to be considered as having crystallized near isotopic
equilibrium with cave dripwater (Hendy, 1971), were performed for each stalagmite. The first
half of the Hendy Test involves analysis of multiple isotopic analyses performed on samples
drilled at increasing distance from the central growth axis along the same series of growth layers.
The conceptual justification for this approach is that dripwater, and thus speleothem calcite, $\delta^{18}O$
values should remain constant down the stalagmite flanks because $^{16}O$ preferentially lost to $CO_2$
out-gassing is replenished by $CO_2$ hydration and hydroxylation reactions. Progressive $^{18}O$
enrichment associated with kinetic effects tied to Rayleigh distillation suggests isotopic
disequilibrium. No such consistent trends toward elevated oxygen isotopic ratios are found (Fig.
8), and thus the BG/GCL stalagmites appear to satisfy the first criterion of the Hendy Test.

The second portion of the Hendy Test is based on the degree of covariation of carbon and

oxygen isotopic ratios. Oxygen isotopic ratios of speleothem calcite reflect those of infiltrating
fluids, which are generally close to the $\delta^{18}O$ values of meteoric precipitation, and which, in many
locations, are linked to climate (air temperature, moisture source, seasonality of precipitation, or
rainfall amount (Lachniet, 2009)). Interpreting changes in oxygen isotope composition at
BG/GCL during intervals of profound climatic change such as marked the last glacial period is
complicated by the multiple factors that influenced $\delta^{18}O$ values of precipitation at these sites,
including shifts in moisture source. The potential exists for rainfall in Iberia to be derived from
atmospheric moisture sources that change on synoptic/seasonal scales (Moreno et al., 2014;
Gimeno et al., 2010; Gimeno et al., 2012) as well as in response to changing glacial boundary
conditions (Florineth and Schlüchter, 2000; Kuhlemann et al., 2008; Luetscher et al., 2016). In
addition, strong but opposite correlations exist in modern precipitation between rainwater $\delta^{18}O$
values and (i) the regional air temperature (r=+0.8) and (ii) rainfall amount (r=-0.8), both of
which are related to the strong seasonality of precipitation associated with Meso-Mediterranean
climates (IPMA, 2016).

Correlations between carbon and oxygen isotope ratios are presented in Figure 8. Three

stalagmites – BG6LR, BG66, and BG67 – show strong correlations between $\delta^{13}C$ and $\delta^{18}O$
($r^2$=0.6), while the other three samples lack a strong correlation. If one considers the second
criterion of the Hendy Test, the nature of equilibrium crystallization in stalagmites BG6LR,
BG66, and BG67 would be considered suspect. It must be noted, however, that the reliability of
the Hendy Test has been questioned because (1) equilibrium may be maintained in some portions
of a stalagmite but not others, (2) growth layers thin progressively down the sides of the
stalagmite, making it difficult to restrict samples to the same material, and (3) equilibrium
covariation of carbon and oxygen isotope ratios may result as the direct or indirect result of
climatic variability (Dorale and Liu, 2009; Lechleitner et al., 2017). We therefore interpret both
isotope ratios and their covariation as environmental signals.

*4.4 Hydroclimate Proxies*
*4.4.1 Carbon Isotopes*

Interpreting speleothem $\delta^{13}C$ variability in a climatic context requires understanding, or

at least constraining, the origins of these isotopic shifts. Stalagmite $\delta^{13}C$ values reflect two
primary inputs: $CO_2$ derived from the atmosphere and/or soil zone and bicarbonate derived from
dissolution of bedrock carbonate. Speleothem $\delta^{13}C$ values reflect the type ($C_3$ vs $C_4$) and density
of vegetation over the cave, both of which are impacted by changes in air temperature and/or
precipitation. The average $\delta^{13}C$ value of biogenic $CO_2$ in the soil zone is tied to the ratio of
plants utilizing the $C_3$ (average $\delta^{13}C$ -26‰) versus $C_4$ (average $\delta^{13}C$ -14‰) photosynthetic
pathways (Deines, 1980; von Fischer et al., 2008). Similarly, vegetation density and soil
respiration rates over the cave impact the relative contribution of atmospheric $CO_2$ (pre-
Industrial $\delta^{13}C$ -6‰ to -7‰; Francey et al., 1999) as compared to soil-derived $CO_2$ (Hellstrom
and McCulloch, 2000; Genty et al., 2003). Phanerozoic bedrock $\delta^{13}C$ values range from -4‰ to
+8‰ (Saltzman and Thomas, 2012), but these values are static and do not contribute to temporal
variability in stalagmite carbon isotopic ratios.

Superimposed on these inputs are secondary effects capable of influencing the $\delta^{13}C$

values of dripwater in the epikarst or cave. When voids in the bedrock are not fully saturated,

$CO_2$ degassing from infiltrated water may occur in the epikarst. This preferential loss of $^{12}CO_2$

(that may result in crystallization of calcium carbonate – so-called prior calcite precipitation)

enriches the residual solution in $^{13}C$, a signal that can be transferred into underlying stalagmites

(Baker et al., 1997). Once the solution enters the cave, equilibrium fractionation between

dissolved carbon species may be disrupted owing to issues surrounding $CO_2$-degassing under

low drip rate conditions (Breitenbach et al., 2015) or by disequilibrium processes occurring

during carbonate crystallization (Mickler et al., 2004; Fairchild et al., 2006). Importantly, $\delta^{13}C$

values reflect local infiltration rather than (pan-)regional atmospheric conditions as in the case of

$\delta^{18}O$. This difference between both proxies offers the opportunity to investigate environmental

changes at different spatial scales.

Terrestrial deposits preserving pollen spectra spanning substantial portions of the last

glacial cycle from western Iberia are rare (Gómez-Orellana et al., 2008; Fletcher et al., 2010;

Moreno et al., 2012), and thus pollen in marine sediments represents a particularly important

continental climate record. Pollen samples obtained from the Iberian margin contain small

percentages of *Poaceae*, the family including the majority of $C_4$ plants, demonstrating a

persistent and overwhelming majority of $C_3$ (largely shrub and arboreal) vegetation throughout

the last glacial cycle including between Greenland stadials (GS) and interstadials (GI) and across

Heinrich stadials (HS) (d'Errico and Sánchez Goñi, 2003; Tzedakis et al., 2004; Desprat et al.,

2006; Sánchez Goñi et al., 2008; Sánchez Goñi et al., 2013; Margari et al., 2014). In the absence

of changes in vegetation type, shifts in the source of carbon found in cave dripwater therefore

likely originated with the density of vegetation and/or soil respiration rates (Genty et al., 2003).

Reductions in these values are generally associated with decreases in temperature and/or

increases in aridity, such as have been inferred from Iberian pollen spectra to have characterized

Iberia during GS, HS, and glacial maxima (Sánchez Goñi et al., 2008; Margari et al., 2014).

Complementing these effects are increases in the contribution of bedrock carbon, as well as prior

calcite precipitation, reflecting a combination of longer residence times of infiltrating solutions

and desaturation of voids in the epikarst above the cave, both of which are consistent with more

arid climates (Baker et al., 1997; Genty et al., 2003). Thus, we interpret the carbon isotopic

values of the BG/GCL record as primarily a local (hydro)climate proxy, with higher $\delta^{13}C$ values

indicative of a cooler, drier climate. Integrating the GCL6 $\delta^{13}C$ record into the BG time series is
complicated by the slightly different bedrock $\delta^{13}C$ values of the host rocks (Supp. Table 1) and
what may have been distinct vegetation types and cave hydrologies at each cave when GCL6
was being deposited (187-160 ka). However, similar $\delta^{13}C$ values during their period of overlap
(187-185 ka) suggests that the two records can be consolidated (see below).

A test of equilibrium crystallization in the modern system can be constructed by

comparing modeled stalagmite isotopic values to recently deposited calcite. The carbon isotopic
composition of speleothem calcite is the result of a complex series of reactions that have been
addressed in a number of studies (Hendy, 1971; Mühlinghaus et al., 2007; Dreybodt, 2008). For
$\delta^{13}C$ in BG stalagmites, we use the equations of Li et al. (2014), which factor in the two primary
sources of carbon – soil $CO_2$ and bedrock carbonate – the proportion of carbon derived from
each source, and temperature-induced fractionation of carbon isotopes between dissolved carbon
species:

$\delta^{13}C_{calcite} = f_1 * [\delta^{13}C_{ls} - (\delta^{13}C_{CO2(g)} + 9.48\text{x}10^3/T - 23.89)] + \delta^{13}C_{CO2(g)} + 9.48\text{x}10^3/T + 0.049T - 37.72$

where: $f_1$ = fraction of bicarbonate from limestone (ls)

T = temperature (˚K)


We assume the most straightforward and simple situation: the system remains closed to

soil $CO_2$ after entering the epikarst and bedrock carbonate contributes 50% of carbon to
dripwater bicarbonate ($f_1$=0.5). We apply the average cave temperature of 14.4˚C and the
measured $\delta^{13}C$ values of BG bedrock and the overlying vegetation/soil of +3±1‰ and -28±1‰,
respectively. This approach, while certainly overly simplified for the BG cave system, yields
modeled stalagmite $\delta^{13}C$ values averaging -7.7±1‰, similar to calcite crystallized on two glass
slides installed at the site of two actively growing stalagmites in the loft area of BG, which
yielded $\delta^{13}C$ values of -8.4±1.2‰.

*4.4.2 Oxygen Isotopes*

The origins of BG/GCL isotopic variability appear more complex for oxygen than for

carbon. Like $\delta^{13}C$ values, local $\delta^{18}O$ minima mark interstadials and interglacials. Analysis of
modern precipitation data reveals equally strong, albeit inverse, correlations between
precipitation $\delta^{18}O$ and both amount (r=-0.8) and air temperature (r=+0.8) effects, likely owing to
the dominance of cool season precipitation in annual water budgets (IAEA/WMO, 2016) (Fig.
2). Based on these relationships, it remains possible that changes in air temperature, overall
precipitation, and/or precipitation seasonality could impact the $\delta^{18}O$ values of effective moisture.
That air temperature is likely not a prominent driver of stalagmite oxygen isotopic variability is
supported by two observations, however. First, the slopes of the air temperature/$\delta^{18}O$
relationships (‰/˚C) at the three GNIP stations located closest to BG and GCL (Porto, Vila Real,
and Portalegre) are nearly identical (average for the three sites of 0.25±0.03‰/˚C) but opposite
in sign to the calcite-water temperature dependence of oxygen isotopic fractionation (-0.2‰/˚C)
(Kim and O'Neil, 1997) (slopes of precipitation amount/$\delta^{18}O$ are -1.6, -3.5, and -3.7‰/100
mm/month, respectively). In the simplest sense, therefore, a 1˚C increase in mean annual air
temperature (and thus also cave temperature) would increase precipitation $\delta^{18}O$ values by
approximately the same amount that the water temperature effect would lower stalagmite calcite
$\delta^{18}O$ values. In this simplified scenario, the net effect is a stalagmite record that is negligibly
influenced by multi-decadal/centennial-scale temperature changes alone. Secondly, the observed
shift toward lower stalagmite $\delta^{18}O$ values during interstadials and interglacials, periods of
elevated mean annual temperature, demonstrates that the observed positive correlation between
precipitation $\delta^{18}O$ and air temperature is not a dominant feature over millennial time scales. For
example, the 3.5‰ decrease in $\delta^{18}O$ values between MIS 6 and MIS 5e (136-128 ka) (Fig. 9) can
be only partially accounted for by the ~1‰ ice volume-related decrease in North Atlantic surface
water $\delta^{18}O$ values (Schrag et al., 1996). Other factors such as kinetics associated with humidity
and wind speed at the point of evaporation (Grootes et al., 1993), temperature and source of
atmospheric moisture (Herbert et al., 2001), and cloud evolutionary pathways (Rozanski and
Araguás, 1995) need also be considered but cannot account for the entirety of this shift. Because
of the narrow continental shelf in central Portugal, the LGM shoreline was located close to the
modern shoreline, thereby minimizing continental effects, and the magnitude of the impacts of
wind speed and ocean temperature do not appear sufficient to account for the observed
stalagmite $\delta^{18}O$ variability. Thus, the decrease in stalagmite $\delta^{18}O$ between the penultimate glacial
and last interglacial suggests that stalagmite oxygen isotope ratios are primarily recording (pan-
)regional hydroclimate rather than temperature. The origin of the anomalously low $\delta^{18}O$ values
during GI 1 (dated here from 14.5-13.9 ka) are unclear (unfortunately no other BG or GCL
stalagmite also spans this interval) but reinforce this inverse relationship between mean annual
temperature and stalagmite oxygen isotope ratios.
Speleothem oxygen isotopic ratios were modeled using the paleotemperature equation of
Kim and O'Neil (1997), which requires measurements of water (cave) temperature and dripwater
$\delta^{18}O$ values. The resulting $\delta^{18}O$ model value of -3.1±1.0‰ is nearly identical to the glass plate-
grown calcite value of -3.0±0.6‰. It should be noted, however, that assessing equilibrium
crystallization in modern calcite/dripwater pairs at BG is complicated by the low temporal
resolution associated with integrated, months-long dripwater samples, variable timing of
dripwater collecting trips, and any seasonal biases in calcite crystallization that at present remain
poorly constrained.
Replication between stalagmites of similar age is arguably the single most reliable
method for evaluating the impacts of climate versus secondary influences, including evaporation
and kinetic effects (Denniston et al., 1999; Mickler et al., 2004), on stalagmite isotopic ratios
(Dorale and Liu, 2009; Denniston et al., 2013). When presented as an integrated data set, the
BG/GCL stalagmite carbon and oxygen isotopic time series spans the majority of the last 220 ka
(Fig. 9), although stalagmites spanning the same periods of time are restricted to 187-185, 111-
104, 83-81, 78-73, and 58-53 ka. Because these intervals are short, and because the temporal
resolution varies substantially between stalagmites, replication tests based on these intervals are
of limited utility. However, within the age uncertainties, $\delta^{18}O$ and $\delta^{13}C$ values and trends are
similar, suggesting that oxygen and carbon isotopic ratios track environmental, rather than drip-
specific, variables. The three exceptions in which coeval samples do not replicate well are: $\delta^{13}C$
values offset by 3‰ from 83-81 ka and by 4‰ from 58-53 ka, and $\delta^{18}O$ values offset by 1‰
from 111-104 ka (Fig. 9; Supp. Fig. S4).

*4.4.3 $\delta^{234}U$ Values*
$\delta^{234}U$ values (calculated as the difference between the age-corrected $^{234}U/^{238}U$ ratio of a
sample and the secular equilibrium $^{234}U/^{238}U$ ratio) of speleothem carbonate have also been used
as a proxy for paleoprecipitation (Hellstrom and McCulloch, 2000; Oster et al., 2012; Plagnes et
al., 2002; Polyak et al., 2012; Zhou et al., 2005). $^{234}U$ exists in the stalagmite crystalline lattice
due to incorporation from cave dripwater and through *in situ* production from decay of $^{238}$U.
Alpha recoil displaces $^{234}$U from its lattice position, increasing its susceptibility to leaching by
infiltrating waters, meaning that $^{234}$U is selectively mobilized relative to $^{238}$U in cave dripwater
(Chabaux et al., 2003; Oster et al., 2012). The flux of infiltrating fluids is therefore tied to $\delta^{234}$U
values of dripwater, and thus stalagmite carbonate, such that decreases in effective precipitation
and/or bedrock dissolution rate, both of which are tied to increased aridity, are associated with
elevated speleothem $\delta^{234}$U values (Hellstrom and McCulloch, 2000; Plagnes et al., 2002; Polyak
et al., 2012).

As differences in $\delta^{234}$U values between stalagmites may arise from distinct infiltration

pathways (Zhou et al., 2005), complicating the integration of $\delta^{234}$U values from multiple
stalagmites into a single, cohesive data set, we restrict our analysis to stalagmite BG6LR, which
represents the longest individual stalagmite record of the BG/GCL time series. While the number
of $\delta^{234}$U measurements is small compared to stable isotopic values, the temporal density of the
former is sufficient to demonstrate the utility of $\delta^{13}$C and $\delta^{18}$O values as paleohydroclimate
proxies (Fig. 9). Decreased precipitation/effective moisture is associated with elevated stalagmite
$\delta^{13}$C, $\delta^{18}$O, and $\delta^{234}$U values. The relationships between $\delta^{13}$C and $\delta^{234}$U values in all BG and
GCL stalagmites are presented in Supp. Fig. S5.

## 5. Environmental Conditions at BG/GCL and Links to Iberian Margin SST

The previously discussed tests for isotopic equilibrium, including the reproducibility of

carbon and oxygen isotope ratios between coeval BG and GCL stalagmites, support the notion
that their $\delta^{13}$C and $\delta^{18}$O values may be integrated into cohesive time series reflecting
paleohydroclimatic conditions and used to assess links between continental climate and SST
(Fig. 10). Over the last several glacial cycles, oceanographic conditions along the western Iberian
margin varied at millennial and orbital time scales in close correlation with Greenland air
temperature and North Atlantic conditions and circulation (Roucoux et al., 2005; Daniau et al.,
2007; Sánchez Goñi et al., 2008; Darfeuil et al., 2016). Abrupt changes in SST reflect a balance
between southward expansion of subpolar waters and northward migration of subtropical water
masses (de Abreu et al., 2003). During the particularly cold conditions characterizing HS and
GS, Iberian margin SST decreased by up to 9˚C (to as much as 13˚C below present values; de
Abreu et al., 2003), with these changes helping to position the arctic or subarctic front at ~39˚N,
the same latitude as BG and GCL. These cold surface waters reduced the production and
transport of atmospheric moisture to Iberia (Eynaud et al., 2009; Voelker and de Abreu, 2011),
and would have thereby influenced the timing of speleothem growth and carbon and oxygen
isotopic values in BG and GCL stalagmites. Indeed, the composite BG/GCL record documents
coherence, at both orbital and millennial scales, between Portuguese hydroclimate, vegetation,
and Iberian margin SST during the last two glacial cycles (Fig. 10 and 11). In an attempt to
quantify this covariance, we binned the SST and stalagmite stable isotope data into century-long
intervals. The relatively short record of BG41 was not included, and model ages for stalagmites
BG66 and GLC6 were increased by 4.0 kyr and 1.3 kyr, respectively, to improve correlation with
the SST chronology. The resulting inverse correlation between SST and carbon and oxygen is
strong (r=-0.55 and -0.52, respectively; p<0.0001) (Supp. Fig. S6).

*5.1 Growth Intervals*
The single most fundamental prerequisite to speleothem deposition is infiltration of
surface waters, and thus the timing of stalagmite growth can reflect changes in mean
hydroclimatic state. Deposition of multiple BG stalagmites was punctuated by hiatuses spanning
similar time intervals (although the precise ages of the onset and/or termination of the hiatuses
are distinct), a relationship that suggests links to changes in hydroclimate rather than random
drip site-specific variability.
Hiatuses in some BG samples coincide with HS1, HS3, HS4, and HS6, and pollen spectra
independently suggest increased aridity during HS and glacial maxima. Decreases in arboreal
pollen abundance and concomitant increases in drought-tolerant vegetation coincide with periods
of reduced SST. Vegetation patterns during maximal IRD deposition on the Iberian margin
reveal not only dramatically reduced forest cover but also a pronounced expansion of semi-desert
plants (e.g. Sánchez Goñi et al., 2000; Roucoux et al., 2005; Naughton et al., 2009). These
changes mark the long hiatus between HS7 and HS6 (71-59 ka), which overlaps with the some of
the coldest SST of the last 70 ka as reconstructed using $U_{37}^{k\,\prime}$ at core MD95-2042 (Darfeuil et al.,
2016) (Fig. 10; Fig. 12). An absence in BG stalagmite deposition from ~160-149 ka occurs at the
same time as massive seasonal discharges from the Fleuve Manche river and the coldest
continental climates and SST (157-154 ka) of the last 220 ka, as determined from pollen and
foraminifera from core MD01-2444 (Margari et al., 2014; Fig. 1).
Whether hiatuses in BG speleothem deposition are a result of pronounced reductions in
precipitation, an extension of below freezing temperatures that limited infiltration (Vaks et al.,
2013; Fankhauser et al., 2016), or variations in infiltration pathway/drip position is ambiguous.
Pollen transfer functions from MD95-2042 suggest winter temperatures dropped below 0˚C
during HS and annual precipitation was reduced by up to 50% (from 800 mm to 500-400 mm
during HS3, HS4, and HS5) (Sánchez Goñi et al., 2002). Applying this temperature
reconstruction to western Portugal is complicated, however, by the broad area across which these
pollen grains were sourced. Permafrost reconstructions (Vandenberghe et al., 2014) of Iberia
argue against the hypothesis that continuous sub-zero temperatures inhibited infiltration and
stalagmite growth. We thus suggest that the hiatuses observed at BG and GCL were driven
largely by reductions in precipitation.
Other western European cave records also share similar growth histories. For example,
stalagmites from Villars Cave, southwestern France (Genty et al., 2003; Genty et al., 2010;
Wainer et al., 2011), and from multiple caves in northern Spain (Stoll et al., 2013) (Fig. 1) are
also punctuated by hiatuses during HS. For example, at or near HS7, stalagmite hiatuses were
formed at Villars Cave (78-76 ka), in northern Spain (~75 k), and BG (80-78 ka). No stalagmite
deposition has been identified at BG from 71-60 ka or Villars cave from 67-62 ka, a period that
includes HS6. Finally, HS1 is marked by a hiatus in northern Spain (18-15.5 ka) and at BG (17-
15 ka). While the timing of these hiatuses is not identical, and not all hiatuses at Villars Cave and
the Spanish caves are coincident with those at BG, the substantial degree of overlap suggests a
common origin. Stoll et al. (2013) noted that stalagmite deposition and/or elevated growth rates
in northern Spain stalagmites occurred during periods of high Northern Hemisphere summer
insolation or during GI, while hiatuses occurred during periods of low insolation and low SST
(<13.7˚C). The BG record supports the hypothesis that growth interruptions are related to SST
controls on regional atmospheric moisture availability, although the impact of insolation is not
clear.

*5.2 BG/GCL Stable Isotopic and $\delta^{234}U$ Variability*
Stalagmite $\delta^{13}C$ and $\delta^{18}O$ values covary with changes in SST at orbital time scales. The
offset between interglacial and glacial isotopic values averages ~3‰ for $\delta^{18}O$ and ~7‰ for $\delta^{13}C$
values (Fig. 10). Stalagmite $\delta^{234}U$ values also preserve these changes in aridity. Millennial-scale
changes are also recorded in stalagmite carbon isotope ratios, with shifts of 3-7‰ associated
with GI/GS transitions, and oxygen isotopic changes of ~1-2‰. The large swing in $\delta^{18}O$ values
during the transition from GI-1 to the Younger Dryas (YD) (~5‰ from 14.0-13.5 ka) is
anomalous. Given that the change in $\delta^{13}C$ values at this time (6‰) is consistent with other GI
transitions, the hydroclimatic implications of this interval require additional study. Similarly,
oxygen and carbon isotopic variability is pronounced during the late Holocene portion of the BG
record. The origin of this high variability is unclear. Replication of the Holocene portion of this
record currently underway will help address this question (Thatcher et al., 2018).
Where growth is continuous during HS, the link between stalagmite isotopic variations
and SST changes is clearly visible (Fig. 11). Prominent positive carbon isotopic excursions
define the YD, HS2, HS5, HS6, and HS8, consistent with diminished concentrations of arboreal
pollen in cores from the Iberian margin, and serve to document particularly cold and dry
conditions at these times (Sánchez Goñi et al., 2000; Roucoux et al., 2006; Sánchez Goñi et al.,
2008). Reduced stalagmite $\delta^{13}C$ values mark periods of enhanced effective moisture from 170-
160 and 145-135 ka, tracking peaks in temperate tree pollen and alkenone-based SST. The BG
record reveals a pronounced increase in stalagmite $\delta^{13}C$ values during the YD, at odds with the
plateau in SST observed in some Portuguese coastal margin sediments at this time. However, a
higher resolution SST record reveals a pronounced drop in SST (Rodrigues et al., 2010), well
matched with the BG isotopic profile and the stalagmite record from Villars Cave.
Hydroclimatic shifts associated with GS and GI are most clearly expressed during MIS
5a and 5b in the BG carbon isotope record (Fig. 11). Other European stalagmite records have
identified GI/GS events from the last glacial period (Genty et al., 2003; Spötl et al., 2006; Boch
et al., 2011; Moseley et al., 2014) (Fig. 10), but the level of resolution recorded in the BG/GCL
time series has not been clearly identified previously in western Iberia. A carbon isotope time
series (albeit with low temporal resolution) of a flowstone from southeastern Spain does not
present clear evidence of either GI or most HS during the last glacial cycle, although it does
contain a clear expression of HS11 (Hodge et al., 2008) (Fig. 1). And while some Iberian lakes
and peat bogs document environmental changes concurrent with HS, no single record, including
one of the longest - the 50 ka time series from the Fuentillejo maar, south-central Spain -
contains a consistent signal for all HS (Vegas et al., 2010; Moreno et al., 2012) (Fig. 1). GS/GI
oscillations during MIS 3 are not clearly defined in BG stalagmites, likely owing to insufficient
temporal resolution, although the BG records do share a resemblance to reconstructed SST
variability (Fig. 11).
Whether the apparent inconsistent linkages between Iberian margin SST and Iberian
hydroclimate are due to the limitations of these proxies, region-specific responses to SST
variations, or a changing influence of SST on precipitation is unclear. However, other points of
divergence between SST and the BG/GCL records exist. For example, some marine cores reveal
a prominent spike in forest taxa occurring at the start of interglacials, decreasing thereafter for
the next 5-10 kyr (Tzedakis et al, 2004; Desprat et al., 2007) (Fig. 10). This early interglacial
peak is a common feature in several time series including the Antarctic δD (Petit et al., 1999)
and $CH_4$ records (Loulergue et al., 2008), and in stalagmite isotopic ratios from the eastern
Mediterranean (Bar-Matthews et al., 2003) and southern France (Couchoud et al., 2009) (Fig.
10). The BG/GCL $\delta^{13}C$ and $\delta^{18}O$ records lack this feature, although the previously discussed
issues surrounding the continuity of the MIS6/5e transition may complicate identifying it.
Stalagmite $\delta^{13}C$ and $\delta^{18}O$ values are lower during GI 20-22 (MIS 5a/4; 84-72 ka) than in
either the Holocene or MIS 5e (Fig. 10 and 12), and BG6LR $\delta^{234}U$ values support this
observation. This interval is of particular interest given that Atlantic forest pollen, which has
been used as a proxy for air temperature, was decoupled from SST across northwestern Iberia
during cold events (C18-C20) (Rousseau et al., 2006; Rasmussen et al., 2014). This decoupling
is interpreted as reflective of a weakened control of SST on Iberian atmospheric temperature
that, in turn, enhanced transport of atmospheric vapor to the high latitudes, amplifying
production of ice sheets in the early stages of the last glacial cycle (Sánchez Goñi et al., 2013).
This process has also been demonstrated for an earlier interglacial (MIS 19; Sánchez Goñi et al.,
2016). Other offsets include (1) the gradual change in BG $\delta^{13}C$ and $\delta^{18}O$ values across the MIS
8/7 boundary, in contrast to the sharp rise in SST at this time; (2) the anomalously large $\delta^{13}C$
response to ice rafting event C24 (111-108 ka), and (3) the persistence of low $\delta^{13}C$ values as SST
decreased from 205-187 ka (Fig. 11 and 12).
The mechanism linking SST and Iberian hydroclimate over millennial time scales
remains unclear. The NAO exerts a strong control over Iberian precipitation, and previous
studies have suggested that GS and GI (Moreno et al., 2002; Sánchez Goñi et al., 2002; Daniau
et al., 2007) and HS (Naughton et al., 2009) were characterized by distinct NAO modes. The
dynamics of the NAO and Azores High pressure system prior to the historical era are only
beginning to be understood (Trouet et al., 2009; Olsen et al., 2012; Wassenburg et al., 2013), and
the BG/GCL record cannot address this question independently. However, rainfall variability in
eastern Iberia is less closely tied to the NAO than is western Iberia and instead reflects other
climatic phenomena including the El Niño-Southern Oscillation (Rodó et al., 1997), helping to
produce an east-west precipitation gradient. Additional high-resolution speleothem records from
central and eastern Iberia could therefore provide a more robust test of the underlying drivers of
millennial-scale hydroclimatic changes during recent glacial periods.

**6. Conclusions**
The BG/GCL composite speleothem record demonstrates that the hydroclimate and
vegetation dynamics in west-central Portugal tracked Iberian margin SST over orbital and
millennial scales during the past two glacial cycles. Enhanced aridity characterized HS, as
evidenced by elevated carbon and oxygen isotopic ratios and/or hiatuses in stalagmite growth,
consistent with other regional stalagmite time series. GI/GS variability expressed in the Iberian
margin SST record and in co-deposited pollen spectra is also present in the BG/GCL time series,
and is particularly well defined in MIS 5a and 5b. Understanding differences between the
structures of the stalagmite and SST records during some time intervals will require development
of speleothem records from central and southern Iberia.

**Acknowledgements**
This work was supported by the Center for Global and Regional Environmental Research
and Cornell College (to R.F.D.), and the U.S. National Science Foundation (grant BCS-1118155
to J.A.H., BCS-1118183 to M.M.B., and AGS-135539 to C.C.U.). Field sampling performed
under the auspices of IGESPAR (to J.A.H.) and Associação de Estudos Subterrâneos e Defesa do
Ambiente. Brandon Zinsious and Stephen Rasin contributed to fieldwork at BG, and Zachary
LaPointe assisted with radioisotopic analyses; Suzanne Ankerstjerne performed stable isotope
measurements. Use of the following data sets is gratefully acknowledged: Global Precipitation
Climatology Center data by the German Weather Service (DWD) accessed through
http://gpcc.dwd.de; NAO Index Data provided by the Climate Analysis Section, NCAR, Boulder,
USA, Hurrell (2003). Updated regularly. Accessed through
https://climatedataguide.ucar.edu/climate-data/hurrell-north-atlantic-oscillation-nao-index-pc-

based. This manuscript benefitted tremendously from discussions with Maria F. Sánchez Goñi, David Hodell, and Chronis Tzedakis. We thank four anonymous reviewers who substantially improved this manuscript's scope and clarity through detailed and thoughtful assessments. Stable and U-series isotope data are available at the NOAA National Centers for Environmental Information website.

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

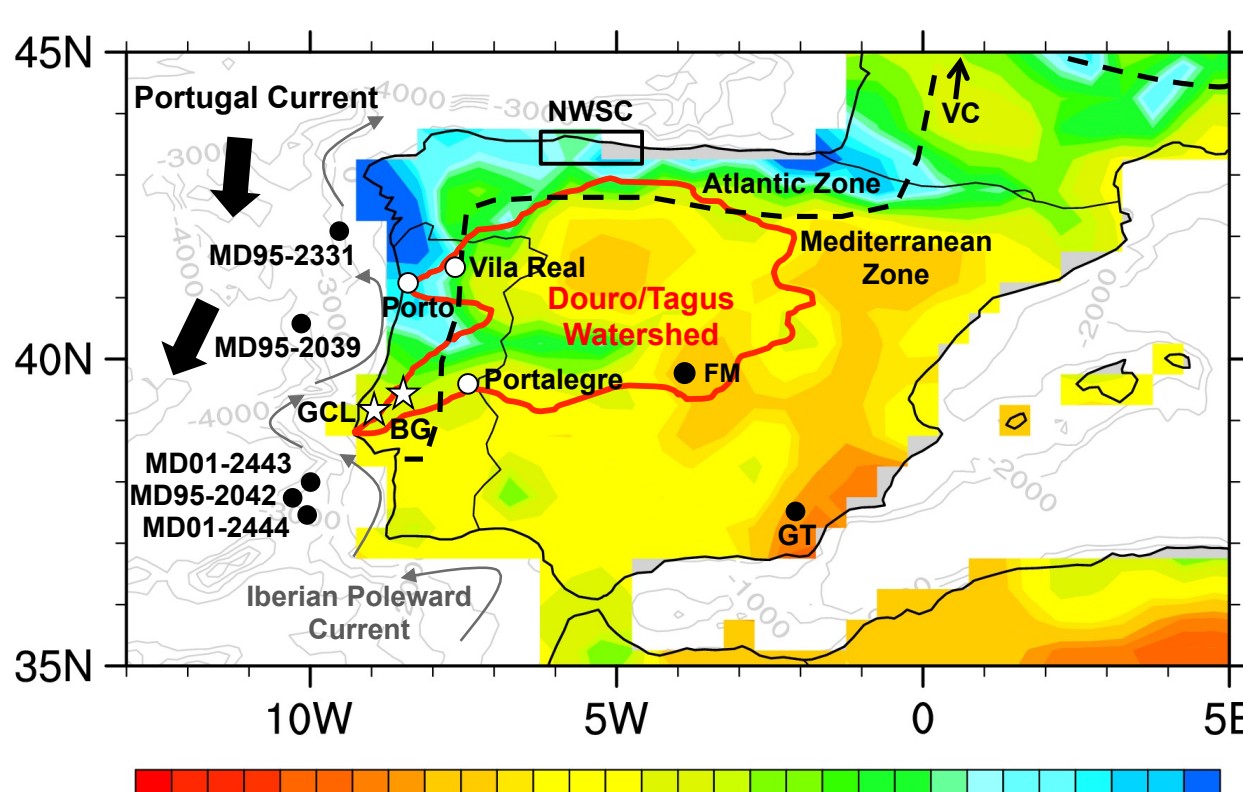


**Figure 1. Average annual precipitation (mm) of the Iberian Peninsula for years AD 1901-2009 (GPCC v. 6; Schneider et al., 2013) relative to cave study sites (white stars: GLC = Gruta do Casal da Lebre; BG = Buraca Gloriosa).** Rectangle denotes location of northwest Spain cave sites (NWSC) (Moreno et al., 2010; Stoll et al., 2013); FM = Fuentillejo maar (Vegas et al., 2010) and GT = Gitana cave (Hodge et al., 2008); VC = Villars Cave (Genty et al., 2003) located just north of map. Also shown are locations of marine cores discussed in text and GNIP stations at Porto, Vila Real, and Portalegre. Bathymetric contours shown in grey (m). Location of currents after Voelker et al. (2010).


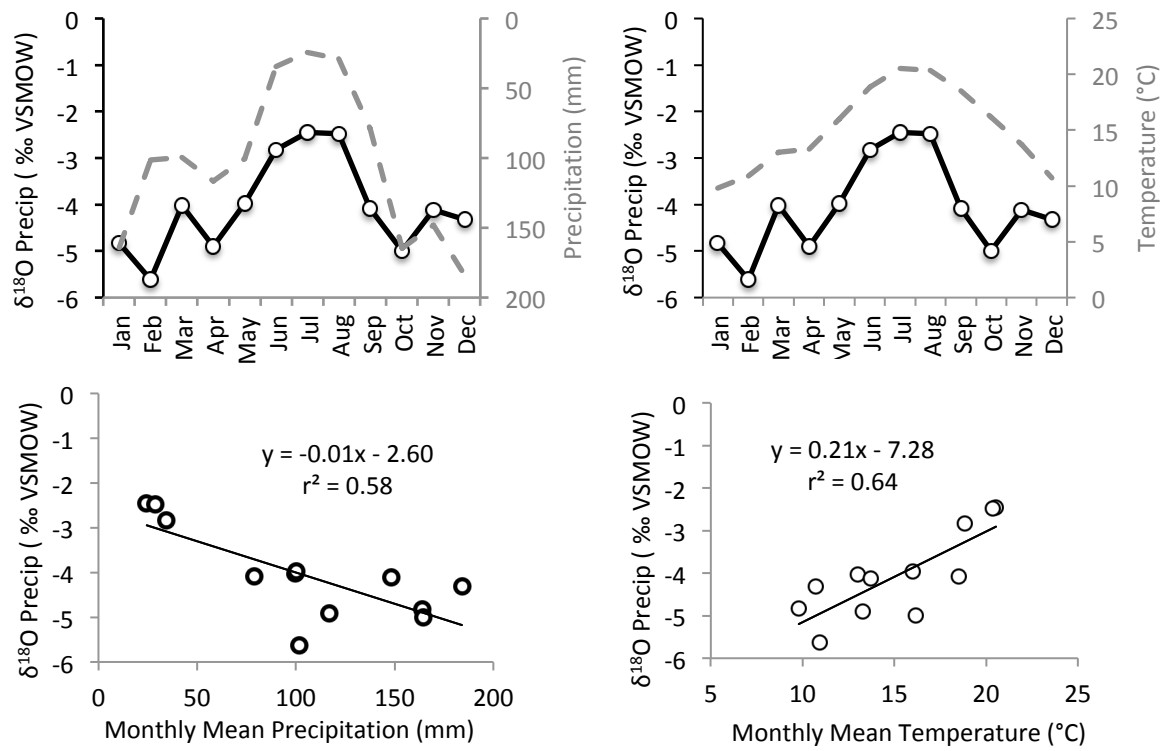




**Figure 2. Oxygen isotopic composition of precipitation versus rainfall amount (lefthand panels) and air temperature (righthand panels).** Data collected at IAEA/GNIP site in Porto, Portugal (see Fig. 1 for location) for 1988-2004. Oxygen isotope data represent multi-year averages of monthly means. The two other closest GNIP stations in Portugal - Vila Real and Portalegre (see Figure 1) - share similar relationships between precipitation oxygen isotopic composition and air temperature (+0.27‰/°C: $r^2$=0.76 and +0.26‰/°C; $r^2$=0.69, respectively) to that of Porto (+0.21‰/°C). The relationship between precipitation oxygen isotopic composition and monthly precipitation amount is -3.5‰/100mm/month ($r^2$=0.64), -3.7‰/100mm/month ($r^2$=0.49), and -1.6‰/100mm/month ($r^2$=0.62) for the three sites, respectively. Note that right hand y-axis in upper left panel is inverted in order to illustrate inverse nature of rainfall and precipitation oxygen isotopic composition.

975

976

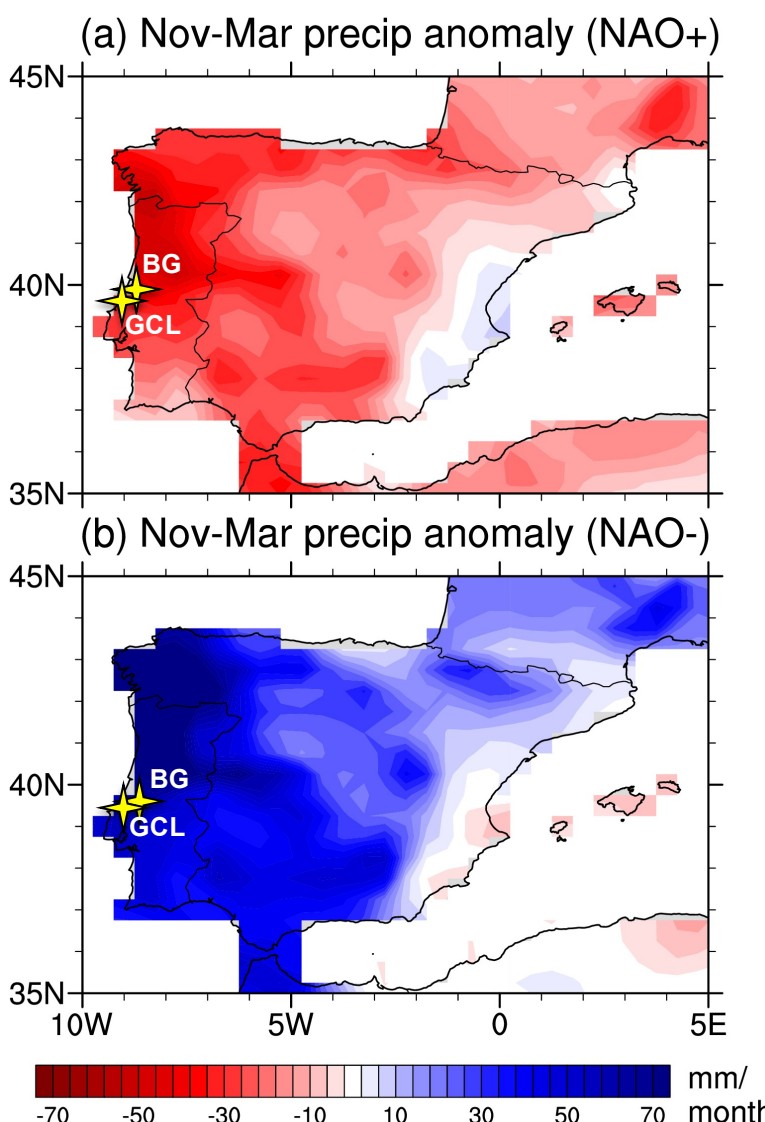

**Figure 3. Iberian rainfall anomalies associated with the North Atlantic Oscillation.** Composites of November-March precipitation anomalies (mm/month) during (a) positive and (b) negative NAO winters for the period 1901-2012. Positive/negative NAO winters were determined using the December-March Hurrell principal component-based NAO index (CDG, 2018) as those winters with NAO values in the highest/lowest decile of all winters. The PC-based NAO index represents the time series of the leading Empirical Orthogonal Function of SLP anomalies over the Atlantic sector, 20°-80°N, 90°W-40°E. Precipitation anomalies are based on the GPCC precipitation, version 7, at 0.5˚ spatial resolution (Schneider et al. 2014). Yellow stars denote cave sites in this study: BG = Buraca Gloriosa; GCL = Gruta do Casal da Lebre.

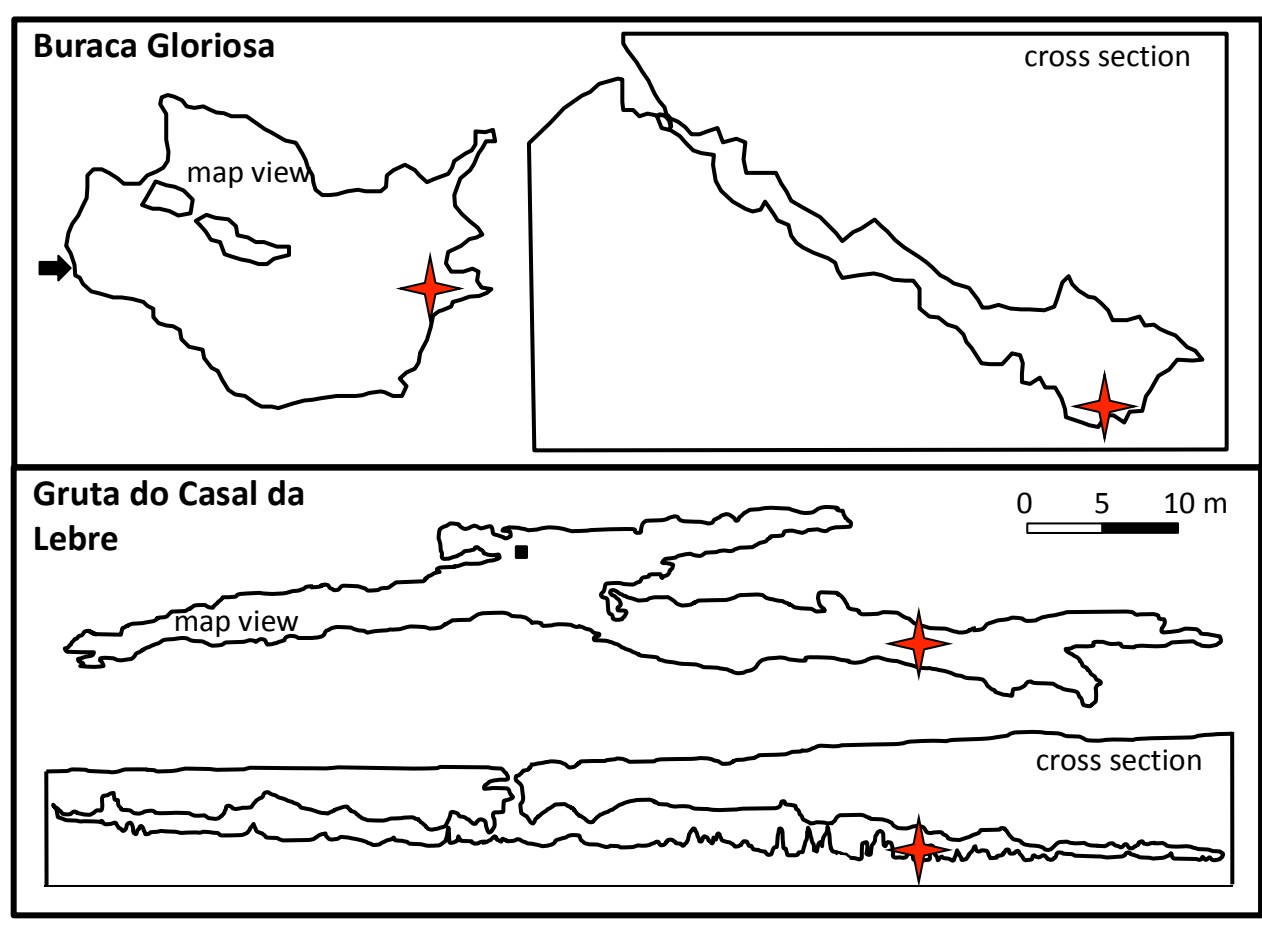

**Figure 4**. Profile and map views of Buraca Gloriosa (top) and Gruta do Casal da Lebre (bottom). Entrance denoted by arrow (top panel) and filled square (bottom panel). Red stars denote locations of stalagmites used in this study.

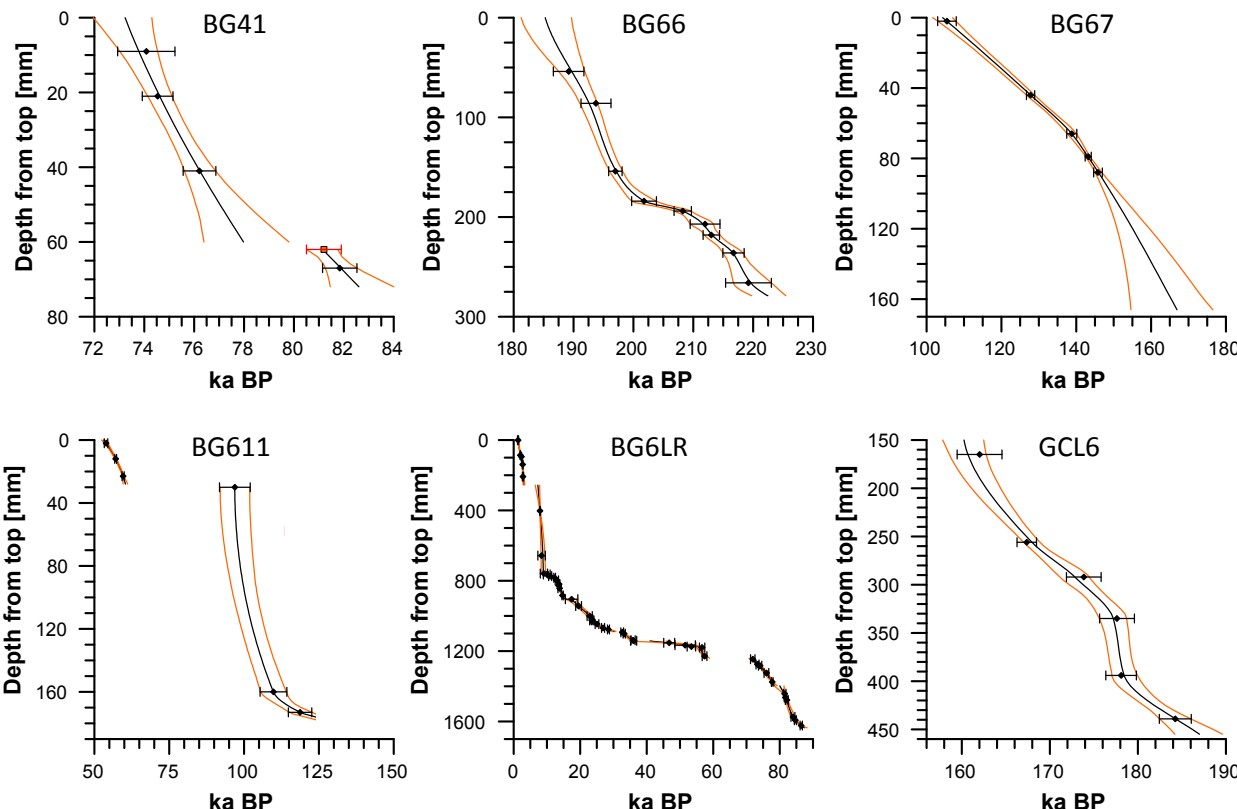

**Figure 5. COPRA-derived age models for BG/GCL stalagmites.** Black lines represent mean of calculated age models while red lines denote 95% confidence intervals. See Table 1 for specific ages and isotopic ratios. Orange square represents a "dummy age" that was included in order to extrapolate below the hiatus, which is only possible with at least two dated points. The bottom of BG611 was based on linear extrapolation through dated intervals. Distances for BG66 were measured relative to topmost section of interval for which stable isotopes were obtained, and not relative to the cap of the stalagmite (see Figure 6).

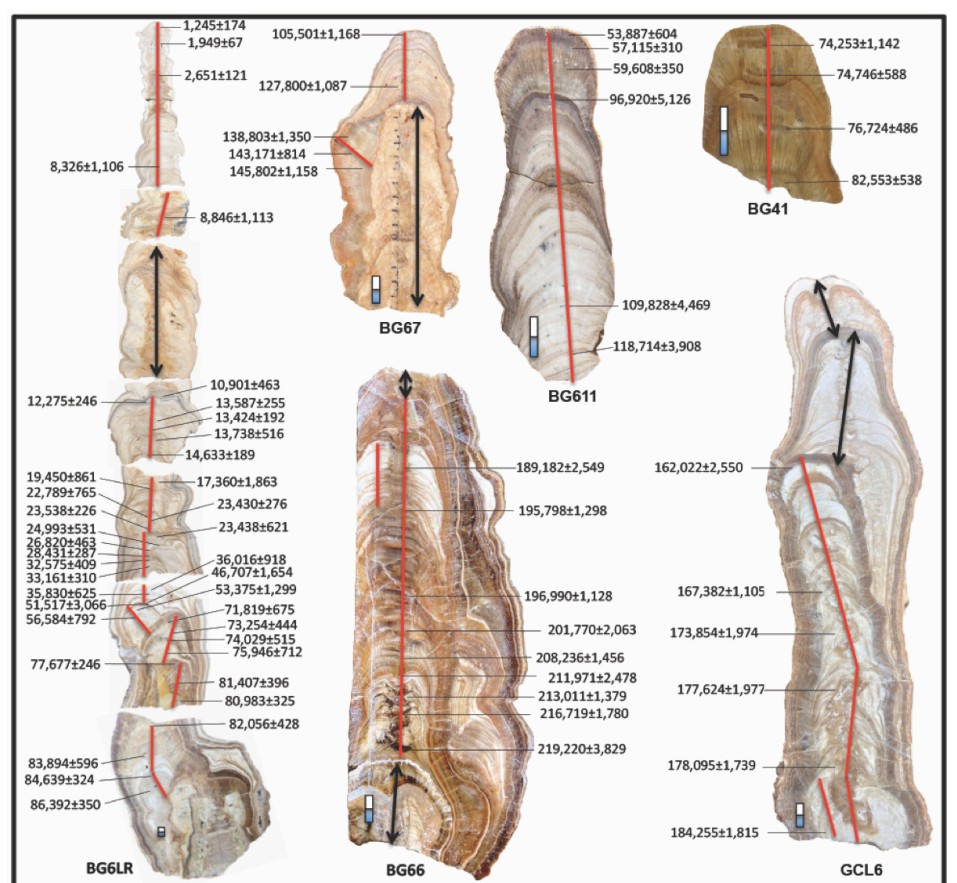

**Figure 6. BG/GCL stalagmites and U/Th ages.** Red lines denote stable isotope sampling transects. Blue and white scale bars (cm) define differential enlargement of each stalagmite. Black arrows represent intervals excluded from this study due to evidence of open system behavior. Sections without arrows or transect lines are older than the interval examined in this study. The impact of recrystallization in stalagmite cores was assessed by parallel sampling transects (parallel red lines on BG66 and GCL6) and demonstrated consistent stable isotopic values and trends (Supp. Fig. S7).

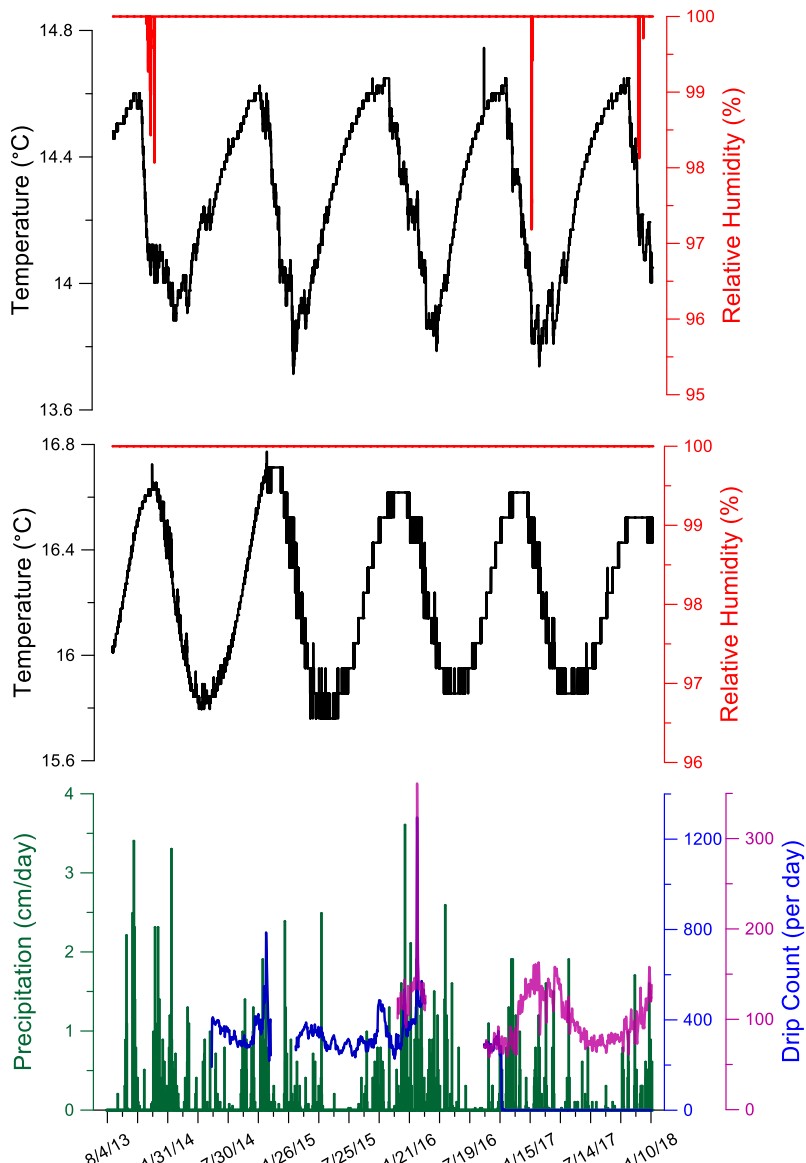

**Figure 7**. Temperature and relative humidity variations from (top) Buraca Gloriosa and (middle) GCL. Drip rate from Buraca Gloriosa and precipitation variability (bottom) from Monte Real, Portugal (35 km from BG). Temperature sensor in GCL was changed in November 2014 and the sensitivity of the new instrument varies slightly from the original.

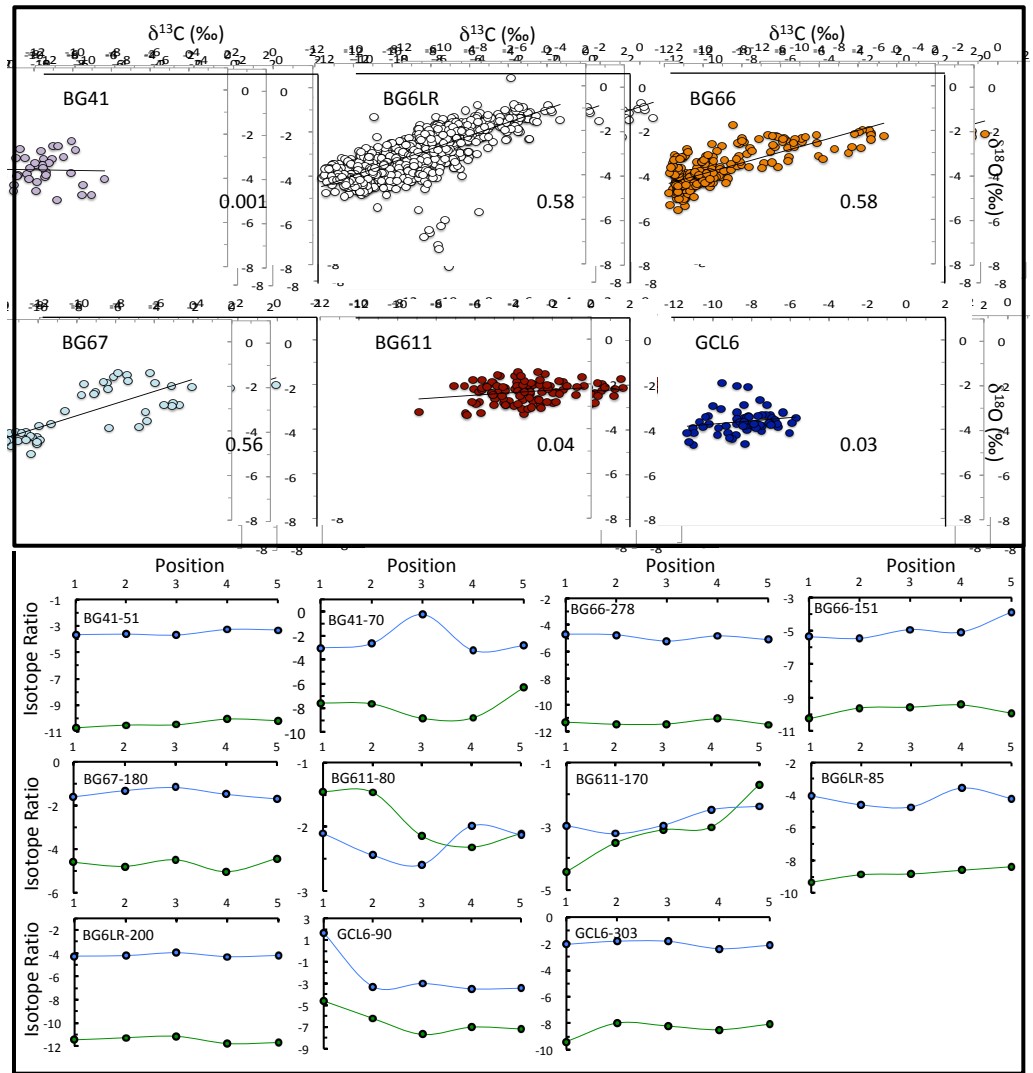

**Figure 8.** Hendy Tests of BG/GCL stalagmites. Top: Covariance plots of carbon and oxygen isotopic ratios. Correlation coefficients ($r^2$ values) are listed for each plot. High positive correlations have been identified as an indicator of non-equilibrium crystallization. Bottom: Oxygen (blue) and carbon (green) isotopic variations along the same growth layers with distance (listed in the upper left corner of each panel) from the stalagmite central growth axis. Progressive increases in $\delta^{18}O$ values have been interpreted to reflect disequilibrium crystallization. Limitations of the Hendy Tests are discussed in text.

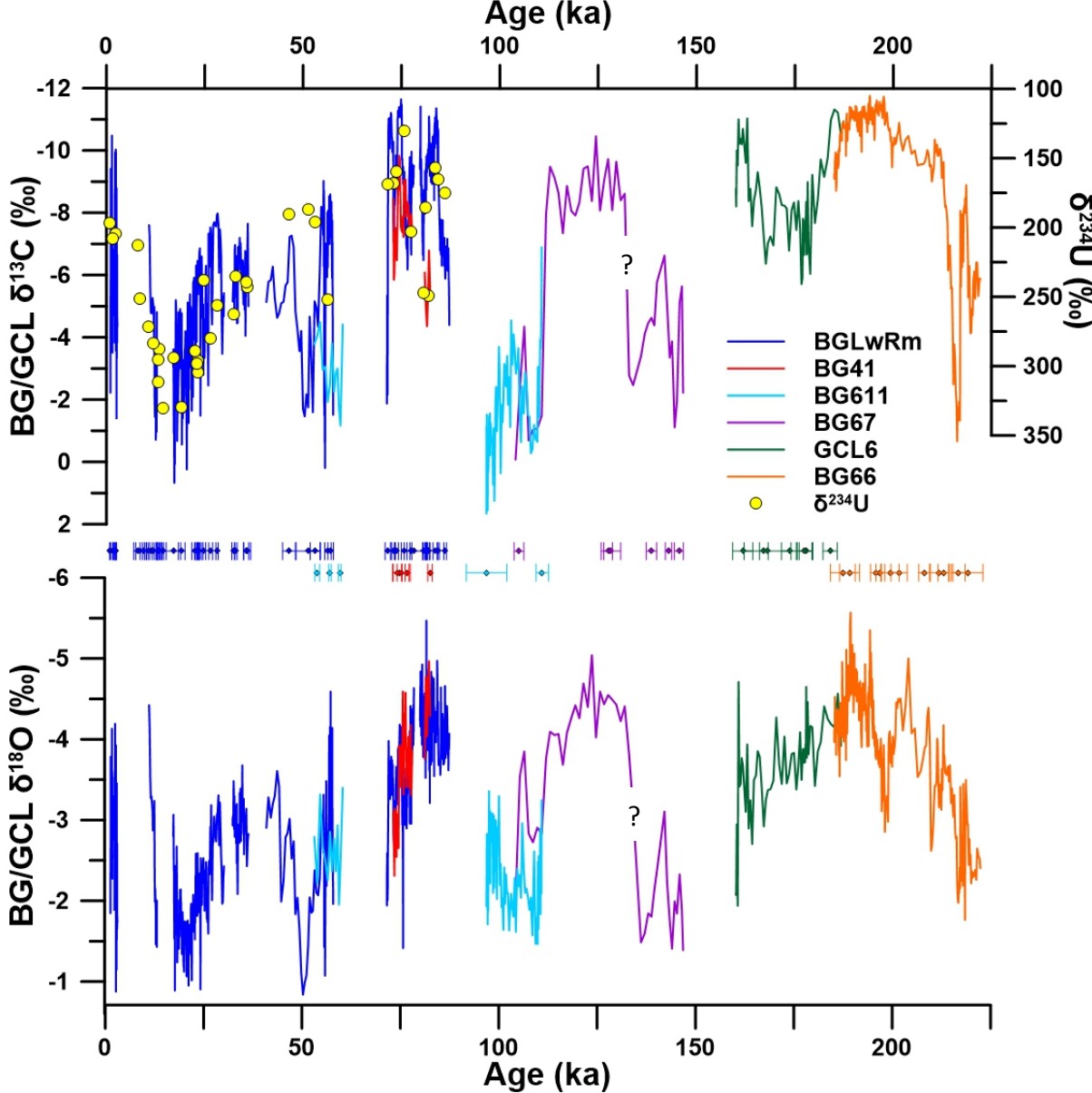

Figure 9. BG/GCL stalagmite isotopic time series. Carbon (top) and oxygen (bottom) isotopes, with each stalagmite presented in a different color. $\delta^{234}U$ values (yellow circles) for BG6LR are plotted against carbon isotope ratios (plots showing the $\delta^{234}U$ and $\delta^{13}C$ values of the other stalagmites are presented in the Supplemental Material). U/Th ages (with 2 s.d. errors) are also shown. The "?" at the MIS 6/5e transition denotes uncertainties associated with the continuity of this interval.

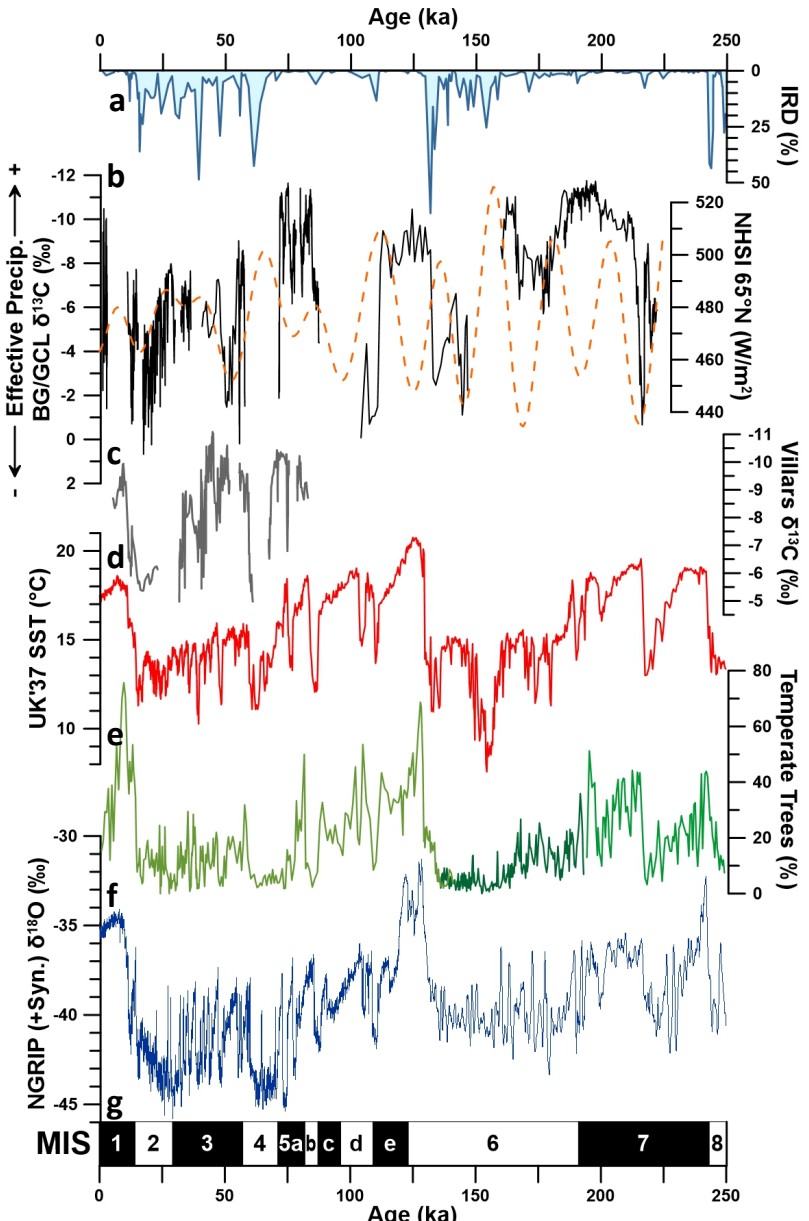

**Figure 10. Comparison of Portuguese stalagmite hydroclimate proxies with regional and**
**global climate records from the last two glacial cycles** (A) Ice-rafted debris abundance from
North Atlantic ODP Site 980 (McManus et al., 1999 using Hulu cave time scale as presented in
Barker et al., 2011); (B) composite BG/GCL stalagmite carbon isotopic time series with NH
summer insolation (Berger an Loutre, 1991); (C) Carbon isotopic time series from Villars Cave,
southern France (Genty et al., 2003; Genty et al., 2006); (D) Alkenone-based Iberian margin SST
reconstruction (core MD01-2443; Martrat et al., 2007); (E) Temperate forest pollen abundance
from three closely spaced cores (MD01-2443: 250-194 ka (Roucoux et al, 2006; Tzedakis et al.,
2004); MD01-2444: 194-136 ka (Margari et al., 2010; Margari et al., 2014); MD95-2042: 136-1
ka (Sánchez Goñi et al., 2008; Sánchez Goñi et al., 2013)); (F) NGRIP (0-122 ka) (North
Greenland Ice Core Project members, 2004) and synthetic Greenland oxygen isotopic record
(Barker et al., 2011) and (G) marine isotope stages.

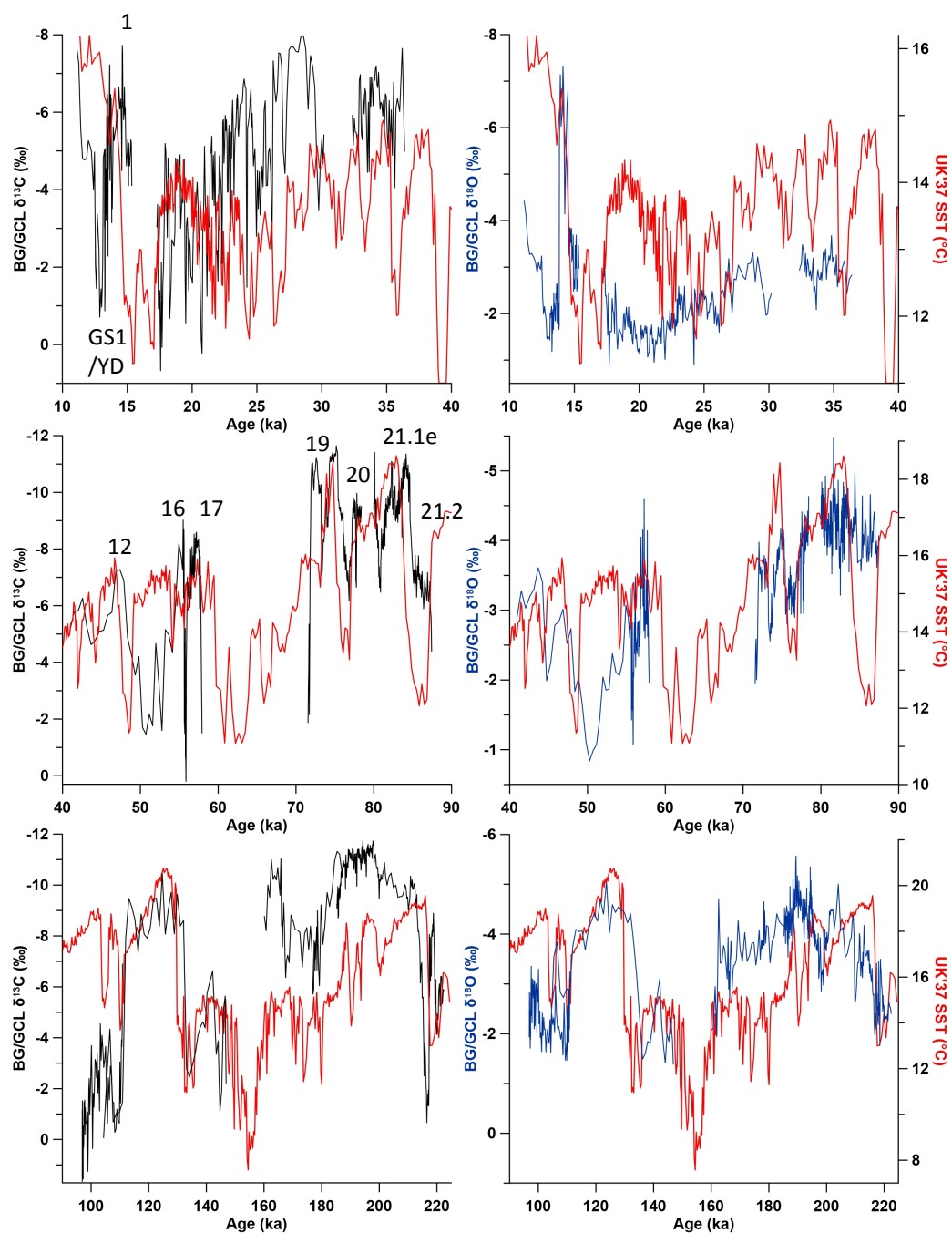

**Figure 11. Iberian margin SST (red) versus stalagmite carbon (black; left column) and**
**oxygen (blue; right column) isotopes.** Numbers denote select GI events using stratigraphic
nomenclature of Rasmussen et al. (2014).

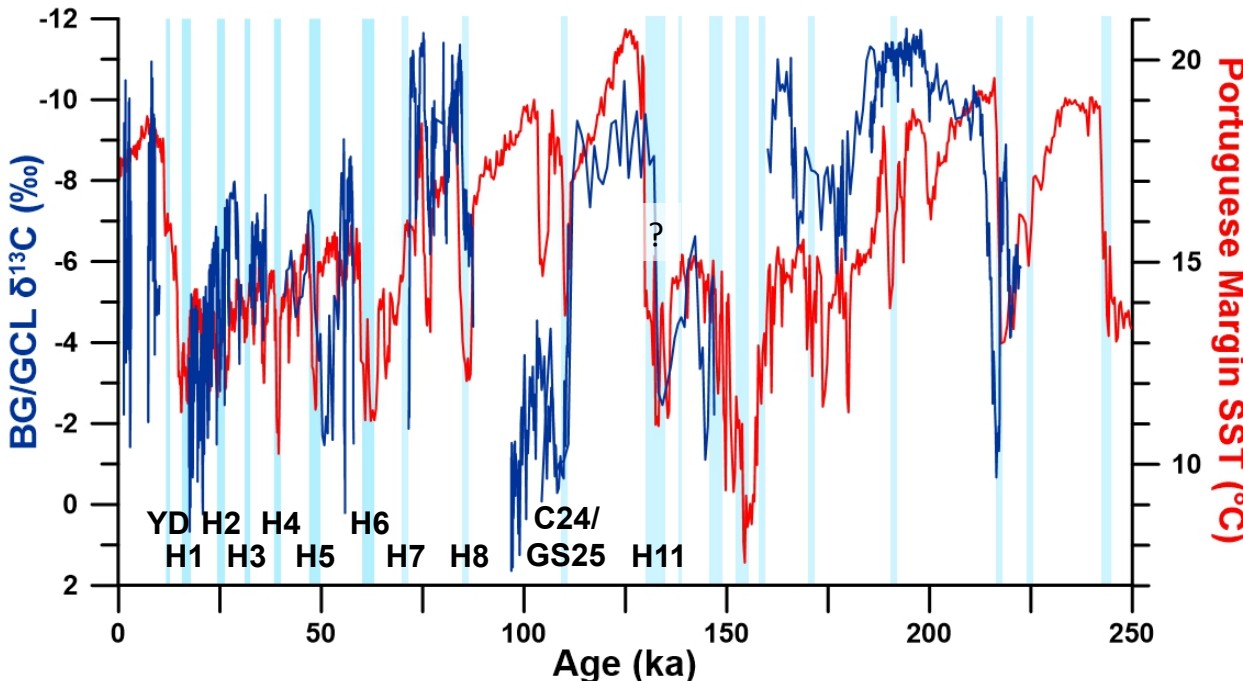

Figure 12. BG/GCL stalagmite carbon isotopic time series and Iberian margin SST. Light blue vertical rectangles denote North Atlantic cold events (some of which are labeled). Several interruptions in stalagmite growth coincide, within the errors of the stalagmite chronologies, with periods of depressed SST. Question mark at MIS6/5e transition denotes visible hiatus not resolvable by U/Th dates.

| Stalagmite | Distance to Top (mm) | $^{238}$U (ng/g) | $^{232}$Th (pg/g) | $\delta^{234}$U[a] (corr'd) | Error[b] | $^{230}$Th/$^{238}$U (activity) | Error | $^{230}$Th/$^{232}$Th (ppm) | Error | Uncorrected Age (yr BP)[c] | Error (yr) | Corrected Age (yr BP)[d] | Error (yr) |
|---|---|---|---|---|---|---|---|---|---|---|---|---|---|
| BG41 | 67 | 148 | 2,892 | 524.7 | 2.2 | 0.779 | 0.0023 | 657.7 | 18.6 | 82,926 | 389 | 82,553 | 538 |
| BG41 | 41 | 293 | 4,635 | 522.8 | 2.2 | 0.742 | 0.0030 | 773.8 | 8.5 | 77,026 | 463 | 76,724 | 486 |
| BG41 | 21 | 217 | 1,858 | 566.6 | 3.1 | 0.748 | 0.0039 | 1,440.0 | 40.6 | 74,906 | 567 | 74,746 | 588 |
| BG41 | 9 | 271 | 2,088 | 610.8 | 9.8 | 0.764 | 0.0073 | 1,635.6 | 22.3 | 74,392 | 1,135 | 74,253 | 1,142 |
| BG66 | 266 | 85 | 6,980 | 698.6 | 9.3 | 1.283 | 0.0057 | 256.5 | 1.8 | 223,637 | 3,252 | 219,220 | 3,829 |
| BG66 | 236 | 123 | 4,742 | 520.6 | 4.1 | 1.169 | 0.0030 | 500.0 | 4.0 | 217,460 | 1,752 | 216,719 | 1,780 |
| BG66 | 218 | 101 | 3,132 | 532.4 | 3.1 | 1.174 | 0.0015 | 623.6 | 4.6 | 214,835 | 1,052 | 213,011 | 1,379 |
| BG66 | 207 | 75 | 4,657 | 429.2 | 3.8 | 1.116 | 0.0025 | 298.1 | 1.7 | 215,891 | 1,580 | 211,971 | 2,478 |
| BG66 | 194 | 68 | 2,003 | 499.5 | 3.1 | 1.149 | 0.0019 | 644.2 | 7.4 | 210,002 | 1,175 | 208,236 | 1,456 |
| BG66 | 184 | 95 | 4,336 | 379.4 | 3.1 | 1.073 | 0.0025 | 386.6 | 3.6 | 204,768 | 1,460 | 201,770 | 2,063 |
| BG66 | 154 | 104 | 2,193 | 443.5 | 2.6 | 1.100 | 0.0015 | 864.2 | 11.9 | 198,297 | 930 | 196,990 | 1,128 |
| BG66 | 86 | 104 | 2,661 | 345.4 | 2.4 | 1.041 | 0.0016 | 672.5 | 8.4 | 197,507 | 994 | 195,798 | 1,298 |
| BG66 | 54 | 76 | 995 | 564.2 | 6.2 | 1.159 | 0.0057 | 1,453.3 | 64.3 | 189,936 | 2,538 | 189,182 | 2,549 |
| BG67 | 88 | 320 | 2,153 | 617.8 | 2.9 | 1.095 | 0.0043 | 2,689.7 | 51.5 | 146,174 | 1,146 | 145,802 | 1,158 |
| BG67 | 79 | 195 | 2,799 | 485.8 | 2.3 | 1.014 | 0.0022 | 1,164.3 | 18.2 | 144,037 | 695 | 143,171 | 814 |
| BG67 | 66 | 250 | 4,187 | 610.3 | 4.9 | 1.072 | 0.0046 | 1,057.4 | 12.7 | 139,735 | 1,279 | 138,803 | 1,350 |
| BG67 | 44 | 162 | 4,858 | 484.7 | 2.4 | 0.969 | 0.0023 | 531.9 | 5.3 | 129,620 | 608 | 127,800 | 1,087 |
| BG67 | 2 | 216 | 5,542 | 401.5 | 2.6 | 0.837 | 0.0039 | 538.0 | 5.1 | 107,150 | 843 | 105,501 | 1,168 |
| BG611 | 173 | 119 | 11,744 | 202.6 | 3.5 | 0.801 | 0.0041 | 133.9 | 0.8 | 126,291 | 1,253 | 118,714 | 3,908 |
| BG611 | 160 | 110 | 12,828 | 230.9 | 4.6 | 0.792 | 0.0044 | 112.1 | 0.7 | 118,672 | 1,277 | 109,828 | 4,469 |
| BG611 | 30 | 122 | 16,801 | 251.3 | 5.0 | 0.762 | 0.0043 | 91.2 | 0.5 | 107,202 | 1,088 | 96,920 | 5,126 |
| BG611 | 23 | 313 | 552 | 340.8 | 1.4 | 0.553 | 0.0024 | 5,168.2 | 353.7 | 59,726 | 345 | 59,608 | 350 |
| BG611 | 12 | 248 | 2,233 | 356.2 | 1.6 | 0.547 | 0.0021 | 1,002.4 | 25.4 | 57,908 | 296 | 57,115 | 310 |
| BG611 | 2 | 250 | 4,109 | 376.7 | 1.8 | 0.533 | 0.0021 | 535.0 | 5.9 | 54,959 | 284 | 53,887 | 604 |
| BG6LR | 1,623 | 72 | 133 | 175.0 | 1.5 | 0.631 | 0.0015 | 5,665.9 | 1,162 | 86,532 | 342 | 86,392 | 350 |
| BG6LR | 1,593 | 98 | 140 | 165.3 | 1.4 | 0.618 | 0.0014 | 7,166.0 | 1,764 | 84,748 | 318 | 84,639 | 324 |
| BG6LR | 1,574 | 74 | 905 | 156.6 | 1.6 | 0.615 | 0.0016 | 824.8 | 25.3 | 84,848 | 360 | 83,894 | 596 |
| BG6LR | 1478 | 159 | 26 | 249.2 | 1.8 | 0.645 | 0.0021 | 63,745.2 | 114,070 | 82,068 | 428 | 82,056 | 428 |
| BG6LR | 1464 | 166 | 1,138 | 246.8 | 1.5 | 0.641 | 0.0009 | 1,542.3 | 35.8 | 81,475 | 214 | 80,983 | 325 |
| BG6LR | 1442 | 162 | 77 | 185.4 | 1.4 | 0.634 | 0.6339 | 21,885.5 | 13,015 | 81,442 | 396 | 81,407 | 396 |
| BG6LR | 1375 | 112 | 220 | 202.9 | 1.5 | 0.602 | 0.6016 | 5,064.2 | 652.0 | 77,823 | 234 | 77,677 | 246 |
| BG6LR | 1324 | 120 | 1,908 | 130.2 | 1.4 | 0.566 | 0.5660 | 585.8 | 15.3 | 77,213 | 330 | 75,946 | 712 |
| BG6LR | 1283 | 132 | 1,019 | 159.5 | 2.0 | 0.566 | 0.5659 | 1,213.9 | 71.1 | 74,623 | 422 | 74,029 | 515 |
| BG6LR | 1276 | 105 | 353 | 167.8 | 2.1 | 0.564 | 0.5637 | 2,766.4 | 298.1 | 73,512 | 425 | 73,254 | 444 |
| BG6LR | 1246 | 83 | 1,232 | 168.7 | 1.4 | 0.561 | 0.5613 | 625.8 | 14.2 | 72,957 | 369 | 71,819 | 675 |
| BG6LR | 1179 | 62 | 1,114 | 252.0 | 2.6 | 0.507 | 0.5071 | 464.4 | 15.9 | 57,877 | 465 | 56,584 | 792 |
| BG6LR | 1174 | 77 | 2,544 | 196.0 | 2.2 | 0.474 | 0.4736 | 235.4 | 3.8 | 55,882 | 375 | 53,375 | 1,299 |
| BG6LR | 1166 | 5 | 367 | 187.1 | 2.6 | 0.482 | 0.4821 | 100.4 | 1.6 | 57,644 | 524 | 51,517 | 3,066 |
| BG6LR | 1153 | 81 | 3,460 | 190.7 | 2.2 | 0.433 | 0.4331 | 167.2 | 2.3 | 49,960 | 367 | 46,707 | 1,654 |
| BG6LR | 1141 | 52 | 1,159 | 242.6 | 2.8 | 0.359 | 0.3591 | 266.4 | 10.4 | 37,626 | 449 | 36,016 | 918 |
| BG6LR | 1138 | 55 | 750 | 239.5 | 1.8 | 0.352 | 0.3518 | 426.3 | 33.1 | 36,815 | 381 | 35,830 | 625 |
| BG6LR | 1101 | 71 | 283 | 235.2 | 2.0 | 0.323 | 0.3234 | 1,344.2 | 198.7 | 33,449 | 272 | 33,161 | 310 |
| BG6LR | 1093 | 70 | 472 | 262.1 | 2.1 | 0.327 | 0.3269 | 802.0 | 73.4 | 33,052 | 331 | 32,575 | 409 |
| BG6LR | 1077 | 101 | 595 | 256.6 | 1.8 | 0.290 | 0.2899 | 810.6 | 63.4 | 28,851 | 193 | 28,431 | 287 |
| BG6LR | 1068 | 85 | 1,034 | 280.0 | 1.4 | 0.285 | 0.2847 | 384.9 | 15.5 | 27,675 | 178 | 26,820 | 463 |
| BG6LR | 1046 | 56 | 705 | 238.2 | 2.2 | 0.260 | 0.2603 | 339.0 | 19.7 | 25,911 | 265 | 24,993 | 531 |
| BG6LR | 1026 | 123 | 2,093 | 304.1 | 1.9 | 0.262 | 0.2617 | 253.3 | 8.5 | 24,612 | 206 | 23,438 | 621 |
| BG6LR | 1025 | 123 | 493 | 296.4 | 1.4 | 0.253 | 0.0017 | 1,041.2 | 151.0 | 23,814 | 175 | 23,538 | 226 |
| BG6LR | 1019 | 80 | 377 | 298.5 | 2.1 | 0.252 | 0.2525 | 887.3 | 107.4 | 23,753 | 221 | 23,430 | 276 |
| BG6LR | 1001 | 68 | 1,464 | 288.7 | 1.5 | 0.256 | 0.2558 | 196.1 | 4.3 | 24,291 | 156 | 22,789 | 765 |
| BG6LR | 944 | 76 | 1,896 | 329.3 | 2.1 | 0.233 | 0.2330 | 154.8 | 3.9 | 21,131 | 196 | 19,450 | 861 |
| BG6LR | 899 | 79 | 4,209 | 294.0 | 3.4 | 0.227 | 0.2266 | 70.6 | 1.3 | 21,074 | 283 | 17,360 | 1,863 |
| BG6LR | 883 | 91 | 233 | 330.3 | 2.0 | 0.168 | 0.1684 | 1,082.0 | 213.7 | 14,806 | 165 | 14,633 | 189 |
| BG6LR | 843 | 100 | 1,409 | 287.7 | 4.0 | 0.162 | 0.1623 | 190.4 | 6.7 | 14,718 | 164 | 13,738 | 516 |
| BG6LR | 827 | 103 | 332 | 295.0 | 2.9 | 0.152 | 0.1521 | 783.5 | 116.9 | 13,645 | 154 | 13,424 | 192 |
| BG6LR | 819 | 75 | 491 | 311.6 | 1.4 | 0.158 | 0.1581 | 400.0 | 22.8 | 14,032 | 123 | 13,587 | 255 |
| BG6LR | 783 | 95 | 525 | 283.8 | 2.2 | 0.141 | 0.1406 | 418.7 | 35.3 | 12,661 | 150 | 12,275 | 246 |
| BG6LR | 774 | 107 | 1,351 | 271.4 | 1.4 | 0.130 | 0.1304 | 169.8 | 5.7 | 11,795 | 119 | 10,901 | 463 |
| BG6LR | 759 | 135 | 4,177 | 251.5 | 1.5 | 0.121 | 0.1210 | 64.7 | 1.0 | 11,071 | 117 | 8,846 | 1,113 |
| BG6LR | 657 | 86 | 2,566 | 212.9 | 1.4 | 0.112 | 0.1120 | 62.1 | 0.9 | 10,540 | 96 | 8,326 | 1,106 |
| BG6LR | 139 | 172 | 323 | 204.2 | 1.7 | 0.031 | 0.0010 | 272.6 | 41.0 | 2,790 | 96 | 2,651 | 121 |
| BG6LR | 86 | 155 | 80 | 207.9 | 1.7 | 0.022 | 0.0007 | 720.9 | 312.3 | 1,987 | 62 | 1,949 | 67 |
| BG6LR | 10 | 122 | 43 | 196.7 | 18.9 | 0.014 | 0.0019 | 677.5 | 519.3 | 1,271 | 173 | 1,245 | 174 |
| GCL6 | 439 | 91 | 2,815 | 76.3 | 2.3 | 0.862 | 0.0029 | 461.2 | 9.3 | 185,093 | 1,779 | 184,255 | 1,815 |
| GCL6 | 394 | 86 | 3,009 | 125.7 | 2.0 | 0.881 | 0.0032 | 415.9 | 6.9 | 179,002 | 1,692 | 178,095 | 1,739 |
| GCL6 | 335 | 70 | 4,579 | 82.7 | 3.0 | 0.856 | 0.0029 | 214.9 | 2.3 | 179,406 | 1,794 | 177,624 | 1,977 |
| GCL6 | 292 | 75 | 2,61 | 78.2 | 2.9 | 0.845 | 0.0035 | 481.0 | 9.0 | 174,639 | 1,949 | 173,854 | 1,974 |
| GCL6 | 256 | 116 | 1,019 | 86.2 | 2.2 | 0.836 | 0.0020 | 1,574.3 | 71.8 | 167,617 | 1,102 | 167,382 | 1,105 |
| GCL6 | 165 | 94 | 2,507 | 122.4 | 4.2 | 0.847 | 0.0049 | 526.3 | 13.4 | 162,712 | 2,368 | 162,022 | 2,550 |

[a]  $\delta^{234}$U$_{meas'd}$ = [($^{234}$U/$^{238}$U)$_{meas'd}$/($^{234}$U/$^{238}$U)$_{eq}$-1] x 10$^3$, where ($^{234}$U/$^{238}$U)$_{eq}$ is secular equilibrium activity ratio: $\lambda_{238}/\lambda_{234}$ = 1.0.  Values reported as permil.
[b] Errors are at the 2$\sigma$ level.
[c] Present is defined as the year AD 1950.
[d] Initial $^{230}$Th/$^{232}$Th atomic ratio of 13.5x10$^{-6}$ ± 6.75x10$^{-6}$ used to correct for unsupported $^{230}$Th in BG stalagmites. GCL stalagmites use 4.4x10$^{-6}$ ± 2.2x10$^{-6}$.

1074

