# Peer review of "A Stalagmite Test of North Atlantic SST and Iberian Hydroclimate Linkages over the Last 2 Two Glacial Cycles 3 Rhawn F. Dennistona\*, Amanda N. Houtsa1, Yemane Asmeromb, Alan D. Wanamaker, Jr.c, 4 Jonathon A. Hawsd, Victor J. Polyak<s"

_Climate of the Past, 2017_

## Referee Comment (RC1) · Anonymous Referee #1 · 17 Dec 2017

Rev of the manuscript:‎

By Denniston et al., Iberia stalagmite I reviewed very similar manuscript by the same authors several months ago submitted to ‎QSR. Unfortunately I found that the authors did not respond to the comments. Reading ‎again the manuscript I only can be more critical.‎ The manuscript deals with the link of Iberian Hydroclimate and North Atlantic SST as ‎evident from speleothems d13C and 234U and growth dynamics of stalagmites from two ‎caves in Western Portugal spanning the last 230 ka. ‎ This study presents new speleothems records from Portugal, a unique and important ‎region exploring how the North Atlantic SST influences the pale-

oclimate and environment ‎of the Iberian Peninsula.‎ The speleothems record from this part of the world is important and new and should be ‎published, but the manuscript needs major revisions.‎ However, there is no well-defined structure to the manuscript. There are too many ‎hypothesis and ideas but with no clear background to support them.‎ There are no descriptions of the caves from where the speleothems were sampled. ‎ The correction factor for one cave is the crustal value and for the other is a value ‎determined from the cave drip water, and the difference is substantial. What is the ‎justification to use different correction factor? What can be the reasons, different host ‎rock, soil type, vegetation? Or maybe determining the correction on present-day drips may ‎not be the correct methodology? ‎ ‎ The authors need to put the Figure of the studied speleothems in the text, not in the ‎supplementary material, and indicate the measured ages on the figure, and where the ‎hiatus are. It is important to add petrographic images showing the altered region and ‎regions of hiatus. ‎ The d18O record follows closely the d13C record. The similar pattern suggests that d18O is ‎also reflecting temperature and humidity, or storm track changes. The authors need to ‎elaborate on this, not to conclude that many factors influence d18O and they include a ‎sentence saying that d18O may be influenced by kinetic effects and evaporation.….. If ‎evaporation and kinetics would be a major process why there is a good correlation with ‎d13C. These kinds of sentences need to be properly discussed. Thus although it is correct ‎that many factors influence d18O, it is also true for d13C. The authors measure the isotopic ‎composition of precipitation and cave water, but prefer not to discuss the d18O of the ‎speleothems, this is strange. ‎ Why d234 is only shown for part of the record in Figure 6. ‎ I would like to see on Fig 6, superimposed also the d18O record.‎ It is clear that during the termination MIS6 to MIS 5 and a more coherent discussion is ‎needed, not just hypothesis and suddenly bring d18O to explain seasonal biases. Did the ‎authors performed Hendy test on those speleothems, do verify which of them might have ‎not form in isotopic equilibrium since the repetition test does not work? ‎ The manuscript is rather

confused and a Table showing periods of non-growth can help. ‎ Did the authors take into consideration the error on the ages and age model in the final ‎correlations with other proxies in Figs. 6and 7‎ The authors don't explore the very good and interesting data. The discussion is missing ‎explanation on the correlation between d13C and 234U, and why there are large changes ‎in d13C during sometime intervals for which there are smaller changes in SST and in the ‎percent of temperate trees?‎ As it is written the study of the speleothems record does not add new insight to the ‎understanding of the relationships between SST and the Iberian Hydroclimate Linkages. ‎The authors try to justify the speleothems record rather than explaining what is unique ‎about the record.‎ To summarize, I find the data important and new, but the style, the arguments, the ‎discussion and the introduction to the paper very weak.‎ The manuscript needs major revision before it can be accepted for publication ‎

---

## Referee Comment (RC2) · Anonymous Referee #2 · 18 Dec 2017

I have previously reviewed this manuscript for Geology and I see some improvements from that previous version (particularly in the presentation of age models, Fig. 5, that is now correct) but still many other of my previous concerns are valid now. This manuscript presents major problems, mainly related to the "quality" of presented samples, the lack of necessary information about the studied caves, and the representation of data, which prevent me to accept it for Climate of the Past. I detail below my main concerns and some suggestions the authors should follow to get their work published.

Still, I would like to emphasize the importance of this dataset, covering with several stalagmites from two Portuguese caves almost entirely last 250 ka (although with very

low overlap). I am totally convinced that speleothems can provide the needed information about land-sea correlations in this region and, in this case, the presented record represents an improvement (I agree with the authors), in terms of chronology and sampling resolution, from previous terrestrial records (eg. lake sediments, see Moreno et al., 2012). The organization of the manuscript, specially the Results section, can be also improved with a more systematic order of data presentation.

1.- Chronology. The six speleothems used for this study are complicated samples in terms of growth axis (very variable along the samples), evidences of dissolution, minor and major hiatus, etc.). In fact, the growth of the six speleothems is very discontinuous and thus making difficult the detection of all the hiatus by U-Th dates. I see two possible ways of improving the chronologies that should be carried out by the authors.

• First, more dates are necessary in some stalagmites and, this time, analysing a higher amount of calcite would be desirable (I already pointed out this in my previous review. . . 50-150 mg for U-Th dating is insufficient with samples where U concentration is low as it happens here). Sampling a higher amount is possible and necessary to get more accurate dates. Errors of above 2000 years are common in Table 1 and I think they can be improved.

• Second, I suggest including some petrographic analyses (thin slides) to help on the identification of hiatus. I am not sure if the authors have done that study since it is not shown but comments on the textures and fabrics are made on 146-157 lines. A figure on this issue in the Supplementary material would be desirable.

Additionally, I would like to see a figure with all the age models together (an example is provided in Fohlmeister et al., 2012) to show the intervals that are really replicated. The authors emphasized along the manuscript the good replication of this dataset and I cannot agree with that. They refer to Fig S2 many times to show replication in d13C records. . .. And in that figure it is evident that replication is really minimal (very short periods and not well reproduced patterns). The authors have to focus their interpre-

tations where chronology was better assessed and replicated and be very cautious where the presence of hiatus was not so well replicated. In fact, sentences like "deposition of multiple stalagmites was punctuated by hiatuses of similar time spans..." (lines 181-184) should be avoided (they are not true) and need to be more concrete: I just see one interval where two stalagmites stop growing at the same time, at ca. 100 ka BP. Other example: "The reproducibility of carbon isotope ratios between coeval BG stalagmites argues that their d13C values may be viewed as an integrated time series not substantially impacted by inter-sample isotopic offsets" (lines 223-225). There is also highlighted the coincidence with hiatus in S France and N Spain stalagmites (lines 291-296) and this is not always true (Figure 6).

2.- Present-day cave environment. To my knowledge, this is the first time paleoclimate records from these two caves are presented. Then, it should be mandatory to understand present-day processes that would help to interpret past records. For example, distinguishing if the correlation of hiatus is with cold or with dry events (or both) is important and must be supported by more present-day data. The authors need to understand what is happening today regarding calcite precipitation in the cave. Does it happens the whole year or focused in the rainy season? Is it more abundant during warmer years? In lines 238-239 it is said that "any seasonal biases in calcite crystallization remain poorly constrained". Then, how can they link the data to NAO that is a winter process? I think that some interpretations will be better supported by more monitoring data.

Besides, the switch of 2 per mil in the d13C record from sample CGL6 "for ease of comparison to SST" needs a justification in the text, not a simple note in the figure caption. I do not think such shift in measured values is justified at all without a deeper understanding of the cave environment (soil, host rock, etc).

3. Representation of data. Finally, I also have some concerns on the representation of data versus age. In general, I find too "optimistic" sentences or interpretations in the text that are not always easy to seen in the figures. A good example is Figure 6 that

is used to emphasize the excellent correspondence of d13C from the stalagmites with other records but the scale does not permit to see it!! Examples: how can we see the positive change during the YD (lines 299-300)? How can we see the hiatus at 80-78 ka (lines 292-293)? What about the "effective moisture from 170-160 ka and 145-135 ka? (lines 303-304). Figure 6 needs more ticks in the x-axis to follow the text and some dashed lines or bars to help the reader to find in the figure the events indicated in the text. Regarding representation of data, I also missed some other records that are cited and compared in the text several times, such as Villars cave or many other marine records. Fig. 8 where a zoom is shown for two different intervals would be the place to include those other records. If not, the reader has to go to previous references to compare visually other figures with this new dataset. For the YD, for example, there are many other records available.

Additionally, I have not found in the text any explanation about the representation of pollen data. Is that a combination of records? A stack? How is it made?

And regarding the representation of ice cores, why do not use the "real" ice core for the beginning of the record? The older part can be compared to the synthetic curve, but for the 0-125 ka I suggest to include NGRIP record.

Minor remarks: - line 285 and line 293. Why Fig. 2?? This is certainly a mistake, I am afraid.

- line 119-120: explain the correction you did using cave drip water

- Table S1.There are many reversals not explained in the text.

References cited: Fohlmeister, J., Schröder-Ritzrau, A., Scholz, D., Spötl, C., Riechelmann, D.F.C., Mudelsee, M., Wackerbarth, A., Gerdes, A., Riechelmann, S., Immenhauser, A., Richter, D.K., Mangini, A., 2012. Bunker Cave stalagmites: an archive for central European Holocene climate variability. Clim. Past 8, 1751–1764. doi:10.5194/cp-8-1751-2012

---

## Referee Comment (RC3) · Anonymous Referee #3 · 3 Jan 2018

Denniston et al provide a new and long $\delta$13C and d234ui reconstruction of hydroclimate from two caves in Portugal. The stalagmites are securely dated, but have many hiatuses which may be related to climatic variation. In general, warm conditions are associated with lower $\delta$13C values, suggesting enhanced soil productivity and or decreased prior calcite precipitation, among several hypotheses for the controls on $\delta$13C. Where available, the d234Ui values show similar variations to the $\delta$13C, suggesting the record is one of effective moisture.

I find the paper to be clearly written and well-documented, with a copious degree of reconciliation with the literature that attempts to integrate multiple lines of evidence for

hydroclimatic change over Iberia. The presentation does not require improvements in general.

However, I wonder if this paper would have more of an impact if it were 2/3rds the length and focused primarily on the record at hand, and its $\delta$13C correlation to the marine SST records? The level of detail is appreciated but may detract from what is by all other accounts is a great record.

---

## Referee Comment (RC4) · Anonymous Referee #4 · 11 Jan 2018

General comment:

This paper presents a long discontinuous palaeoclimate record covering the majority of the last 230 ka based on seven speleothems from Portugal. The d13C record shows substantial similarity with SST off the Iberian Margin on orbital to millennial time-scales suggesting that speleothem d13C values reflect past climate variability. It is supplemented by d234U and stalagmite growth resulting in an, overall, consistent picture with more negative d13C and d234U values for periods of warmer SST. In contrast, colder periods, such as the HS and the GS are reflected by hiatuses (in some cases) and more positive d13C and d234U values. In general, this is a nice speleothem record

based on several stalagmites and a large number of U-series ages, and its potential as a climate proxy is evident. Thus, this record should eventually be published.

However, I have several problems with the paper in its current form:

1. Presentation of d18O values: My major concern is that the d18O values are only shown in the supplement although they - as is also acknowledged by the authors themselves - show some similarity with speleothem d13C and SST off the Iberian margin. I agree that the interpretation of speleothem d18O values on these long time-scales may be not straightforward, but the same is true for d13C. Actually, speleothem d18O values should even be less influenced by local or drip site-specific effects than d13C values and show a more consistent signal. Thus, I am convinced that a combined presentation and discussion of the d13C and the d18O values (similarities/differences) would be much more informative for the reader and also result in a more robust climate record. 2. Discussion of d13C values: The discussion of the d13C data in terms of climate variability is by far too short. The authors mention several processes potentially affecting speleothem d13C values, but then do not discuss which of these processes they consider most important for the observed orbital and millennial scale variability of their record. Is it vegetation density and, resulting from that, soil pCO2? Is it the degree of PCP? Is it drip rate and the resulting changes in disequilibrium isotope fractionation? Or a combination of all these processes? As mentioned above, a detailed comparison (maybe for individual stalagmites) with the d18O records could provide additional information. Even if it is not possible to identify one or two dominant processes, the discussion must be extended in order to present this important information to the reader. 3. General presentation of the data: The authors state several times in the MS that the d13C (and d18O) signals recorded in the different stalagmites agree in phases of concurrent growth. I agree that this is an important criterion to test whether the stable isotope signals reflects changes in past climate or are dominated by changes in the local karst system/cave. However, in the current form (Figures 6 and S2), it is almost impossible for the reader to judge how good this agreement really is. Please present

the corresponding sections on a different time-scale (in several plots) or maybe even calculate correlation coefficients. This is the only way to clearly present the agreement and the differences in timing, evolution and absolute values. 4. Dating: The total record is based on 76 ages, and I absolutely acknowledge the associated amount of work and costs. However, whereas for some stalagmites (BG6LR), a large number of ages were determined, for others (BG68, BG67, BG41) only a few samples were dated (and even "dummy" ages inserted). As some of the records clearly show hiatuses that are not accounted for by the current age models (BG67, BG68, Fig. 5), I strongly suggest to date a few more samples (10 may be enough) to improve the age models and, in particular, to better constrain the timing of the hiatuses.

In summary, the paper is well written, the data are interesting and have the potential to make an important contribution to reconstruction of terrestrial climate change on the Iberian Peninsula. However, in its current form, I cannot recommend the paper for publication in CP due to the points mentioned above. Further, detailed comments are provided below.

Detailed comments:

Lines 85ff.: The results of the cave monitoring, in particular the detailed drip rate data should be presented in the results section.

Lines 120-121: Please provide more details about the "isotopic analysis of cave drip water" (methods, results, number of samples, etc.).

Line 171: MIS 9 should be MIS 7?

Line 178 ff.: "The similar carbon isotopic trends and values across many of the areas of overlap argue for a consistent climate signal as the primary driver of isotopic variability (Fig. S2)." As stated in my general comment, the similarity is not visible in the current diagrams. Please provide more detailed plots for the corresponding time intervals.

Line 189ff.: "rate of $CO_2$-degassing from water entering the cave" This is a very common mistake in the speleothem literature. The rate of degassing is always very fast. The degree of degassing, however, which is determined by cave pCO2, has an influence on supersaturation and precipitation rate, which in turn may result in disequilibrium effects. It should also be noted that the degree of disequilibrium will be modulated by drip interval, with long intervals resulting in a higher degree of disequilibrium.

Line 199ff.: "Thus, increases in carbon isotopic ratios are interpreted here as primarily reflecting a combination of desaturation of voids above the cave and decreased organic CO2 production within the soil zone, both of which are consistent with a vegetative response to cooler, more arid climates (Baker et al., 1997; Genty et al., 2003)." Although I generally agree with the interpretation of the authors that more positive d13C values probably reflect drier conditions, this conclusion requires more discussion of other potential processes (see my general comment). The only process that is reasonably excluded are changes in vegetation type (C3/C4) based on pollen evidence. All other processes (changes in drip rate, supersaturation, ageing of organic material in the soil, etc.) are mentioned, but not discussed at all.

Line 207ff.: "Decreases in effective precipitation and/or bedrock dissolution rate, both of which are associated with increased aridity, have been tied to elevated speleothem $\delta$234U values (Hellstrom and McCulloch, 2000; Plagnes et al., 2002; Polyak et al., 2012), and are interpreted similarly here." Even if this has been discussed elsewhere, for the interested reader, it would be good to briefly (2-3 sentences) mention the underlying process here.

Line 211ff.: "As differences in $\delta$234U values between stalagmites may arise from distinct infiltration pathways, we restrict this part of the analysis solely to stalagmite BG6LR, which represents the longest individual stalagmite record of this time series." The same argument holds true for d13C values, which may also strongly depend on differences in infiltration pathways. Differences between the individual stalagmites (in agreement between d13C and d234U) may even provide additional information about the processes occurring in the karst. Please show all the d234U records.

Line 215ff.: Even if the interpretation of the d18O values may be difficult, I am convinced that they contain important information, which should be presented to the reader and discussed in detail (see general comment).

Line 231ff.: "The consistency and coherence among carbon (and oxygen) isotope values of coeval stalagmites . . ." Again, this must be presented in a more comprehensive and quantitative way. In the current form, the reader simply has to believe this statement.

Line 233ff.: ". . . the most notable of which is the shift toward higher $\delta$234 C values at the MIS 5e/6 transition ($\sim$130 ka) in stalagmite BG611 that contrasts with the sharp decrease in carbon isotopic ratios in BG67 (Fig. 6)." The corresponding growth phase in stalagmite BG611 appears very short to me (just a few stable isotope data points). Thus, I would not give too much weight to this section of the record. This again highlights the necessity to present the data on a different age scale better showing the details of the record.

Line 264: HS11 should be HS6?

Line 330ff.: "This early interglacial peak . . ." Please highlight the corresponding feature in Fig. 7. It is not clear to me which peak is meant.

Line 333: Fig. 2 should be Fig. 1?

Line 335ff.: "Next, stalagmite $\delta$13C values are lower during GI 20-22 (MIS 5a/4; 84-72 ka) than in either the Holocene or MIS 5e, suggesting that maximum warmth and precipitation were not coincident with peak summer insolation ($\sim$127 ka) (Fig. 6)." I strongly disagree with this statement. In particular on these long time-scales, differences in absolute d13C values should not be interpreted in terms of the warmest/ coldest or the driest/wettest period. As the authors acknowledge themselves, a variety of parameters may change on these time-scales (karst properties, vegetation type and density, cave ventilation, etc.). Thus, the absolute values should be interpreted with

caution.

Line 348: "... than expected based on the observed scaling with SST (Fig. 7)." This is an interesting point, which should be extended in a revised version of the MS. How good is this scaling for the whole record? It may be interesting to see a scatter plot of speleothem d13C vs. SST. In this context, how about the relation between MIS 7 and MIS 5? In the speleothem record, MIS 7 exhibits lower d13C values than MIS 5, which is not the case in all other climate records presented in the paper (Figs. 6 and 7).

Line 351ff.: "Alternatively, changes in the nature of the NAO ..." I would remove the whole discussion on the NAO, which appears rather speculative to me. The NAO is an inter-annual phenomenon, and even if some studies have suggested persistent phases of NAO+ and NAO- in the past, a discussion on the millennial or even orbital time-scale is difficult. Furthermore, this would provide more space for a detailed presentation and discussion of the d18O values and the potential processes influencing the stable isotope signals.

Line 381ff.: "Differences between the structure of the stalagmite and SST records during some time intervals suggest that land-sea connections across Iberia may have varied temporally and spatially." This statement goes too far (see above).

Line 667ff.: "Conservative errors were added to account for the unknown "true" age of the stalagmite at these points." What do you mean by "conservative" errors? Please explain and motivate in detail how those were defined.

Fig. 3: Due to the long residence time of the water in the aquifer above the cave, the d18O signal of precipitation is smoothed (at least to some extent). Thus, instead of monthly means, it would be better to show the inter-annual variability and relationships.

Fig. 4: Rather than showing just one year for NAO+ and NAO-, it would be better to show a mean state of all NAO+ and NAO- years within a specific period (e.g., the last 50 years).

Fig. 5: Check labelling of the plots. Some speleothem names are different than in the text. In addition, it appears to me that some of the samples contain apparent hiatuses (e.g., BG67), which are not resolved by the current dating and, thus, not accounted for by the age models. Therefore, I strongly recommend to determine a few more ages to improve the chronologies of these samples and to better constrain the hiatuses.

Suggested additional references discussing climate variability on the Iberian Peninsula and in the Mediterranean as well as the timing of orbital and millennial scale climate change:

Other Spanish speleothem records: Muñoz-García MB, Martín-Chivelet J, Rossi C, Ford DC, Schwarcz HP (2007) Chronology of Termination II and the Last Interglacial Period in North Spain based on stable isotope records of stalagmites from Cueva del Cobre (Palencia). Journal of Iberian Geology 33(1): 17–30.

Rossi C, Mertz-Kraus R, Osete M-L (2014) Paleoclimate variability during the Blake geomagnetic excursion (MIS 5d) deduced from a speleothem record. Quaternary Science Reviews 102: 166–180. doi:10.1016/j.quascirev.2014.08.007.

Fletcher WJ, Sánchez Goñi MF (2008) Orbital- and sub-orbital-scale climate impacts on vegetation of the western Mediterranean basin over the last 48,000 yr. Quaternary Research 70(3): 451–464. doi:10.1016/j.yqres.2008.07.002.

Environmental changes in SE Spain Candy I, Black S (2009) The timing of Quaternary calcrete development in semi-arid southeast Spain: Investigating the role of climate on calcrete genesis. Sedimentary Geology 218(1-4): 6–15. doi:10.1016/j.sedgeo.2009.03.005.

For the transition from MIS 6 to MIS 5: Regattieri E, Zanchetta G, Drysdale RN, Isola I, Hellstrom JC, Roncioni A (2014) A continuous stable isotope record from the penultimate glacial maximum to the Last Interglacial (159–121 ka) from Tana Che Urla Cave (Apuan Alps, central Italy). Quaternary Research 82(02): 450–461.

[Figure]

doi:10.1016/j.yqres.2014.05.005.

MIS 6.5: Bard E, Delaygue G, Rostek F, Antonioli F, Silenzi S, Schrag DP (2002) Hydrological conditions over the western Mediterranean basin during the deposition of the cold Sapropel 6 (ca. 175 kyr BP). Earth and Planetary Science Letters 202(2): 481–494. doi:10.1016/S0012-821X(02)00788-4.

MIS 7: Spötl C, Scholz D, Mangini A (2008) A terrestrial U/Th-dated stable isotope record of the Penultimate Interglacial. Earth and Planetary Science Letters 276(3-4): 283–292. doi:10.1016/j.epsl.2008.09.029.

---

## Editor Comment (EC1) · N. Combourieu Nebout (Editor) · 22 Jan 2018

Dear authors,

We have now received the comment of the reviewers. Both reviews underline the potential of your research and the need to publish speleothem records from the studied area.

Please post your replies to all the comments on the discussion forum that explain how you want to amend your manuscript following the reviewer suggestions. If you post a revised version of your paper accordingly, I would like to see your corrections in track

change mode.

I will give my decision on your document after reading these documents.

Looking forward to your responses. With my best regards

Nathalie Combourieu-Nebout

---

## Author Comment (AC1) · 25 Jan 2018

As is described below, we are currently in the process of obtaining additional information from the caves and the stalagmites described in this study. We are requesting from the editor additional time in order to fully address the comments made by each of the reviewers. Thus, our responses should be considered preliminary and incomplete versions of the detailed and developed discussion that will accompany a later revised draft of this manuscript. Here we present the itemized comments by the reviewer followed by our response.

There is no well-defined structure to the manuscript.

[Figure]

We are adding additional (sub)headings, including sections on the construction of a multi-year cave monitoring protocol and its results, and the potential drivers of oxygen isotopic variability.

There are too many hypothesis and ideas but with no clear background to support them. There are no descriptions of the caves from where the speleothems were sampled.

We are adding to the Supplemental Material maps of both caves. BG has never been mapped but we are currently working with an experienced team of cavers to create a map suitable for publication. A previously constructed map of GCL will be adapted for publication. More detail on the ages and isotopic compositions of the bedrock hosting each cave will be added, as will isotopic compositions of vegetation and soil organic carbon. In addition, we have recently (January 2018) visited both caves and collected additional environmental monitoring data (temperature, barometric pressure, humidity, and drip rates) acquired via data loggers over the past 1-3 years, and dripwater obtained over the recent months.

The correction factor for one cave is the crustal value and for the other is a value determined from the cave drip water, and the difference is substantial. What is the justification to use different correction factor? What can be the reasons, different host rock, soil type, vegetation? Or maybe determining the correction on present-day drips may not be the correct methodology?

The GCL samples contain low $^{232}Th$ abundances and are largely insensitive to the initial Th correction. Whether an initial $^{230}Th/^{232}Th$ ratio of 4.4 ppm or 13.5 ppm is used, the ages are nearly identical. And given that the caves are formed in different bedrock and that we lack independent constraints on the initial Th ratio, we use the default value of 4.4 ppm.

The authors need to put the Figure of the studied speleothems in the text, not in the supplementary material, and indicate the measured ages on the figure, and where the

hiatus are. It is important to add petrographic images Ëİshowing the altered region and regions of hiatus.

This figure will be adjusted as requested and moved from the Supplemental Material to the manuscript.

The d18O record follows closely the d13C record. The similar pattern suggests that d18O is also reflecting temperature and humidity, or storm track changes. The authors need to elaborate on this, not to conclude that many factors influence d18O and they include a sentence saying that d18O may be influenced by kinetic effects and evaporation. . ... If evaporation and kinetics would be a major process why there is a good correlation with d13C. These kinds of sentences need to be properly discussed. Thus although it is correct that many factors influence d18O, it is also true for d13C.

This is a fair point. We mention in the manuscript that changes in carbon isotopic ratios can represent a response to hydroclimate which can be associated with vegetation type or density, changes in soil microbial activity, PCP in the epikarst, shifts in drip rates and degassing of CO2 from water into cave air, and kinetics. As for oxygen, d18O responds to circulation, rainfall amount, air temperature, and seasonality. Thus, during times of "drying", one might expect to see increases in d13C and d18O if rainfall is diminished, rainy season is shorter, moisture source more proximal, or temperatures higher. The discussion integrating both d18O and d13C will be expanded and the d18O data will be included in the figures.

The authors measure the isotopic composition of precipitation and cave water, but prefer not to discuss the d18O of the speleothems, this is strange.

This is a fair point. Please see previous reply.

Why d234 is only shown for part of the record in Figure 6. I would like to see on Fig 6, superimposed also the d18O record.

We attempted to construct a composite figure showing all of the d234U data, but the

image was confused by the offsets between individual stalagmites. The only stalagmite for which the comparison of d234U and d13C made sense was BGLR6, which spans the longest interval of time. We will, however, create a similar figure using the d18O data from this stalagmite.

It is clear that during the termination MIS6 to MIS 5 and a more coherent discussion is needed, not just hypothesis and suddenly bring d18O to explain seasonal biases.

We will expand the discussion of the 6/5e transition and also go into greater depth d18O regarding the origin of oxygen isotopic variability.

Did the authors performed Hendy test on those speleothems, do verify which of them might have not form in isotopic equilibrium since the repetition test does not work?

While we are not convinced that the Hendy Test is a reliable means of assessing equilibrium crystallization (as demonstrated convincingly in Dorale and Liu, 2009), we are performing Hendy tests to address this comment. These analyses are currently underway and the results will be presented in the revised version of the manuscript.

The manuscript is rather confused and a Table showing periods of non-growth can help. Did the authors take into consideration the error on the ages and age model in the final correlations with other proxies in Figs. 6 and 7?

The data were presented based solely on the age models.

The authors don't explore the very good and interesting data. The discussion is missing explanation on the correlation between d13C and 234U.

We will include a more thorough explanation of the links between d234U and d13C in the revised discussion. The d234U data are not meant to represent the same sort of fine-scale paleohydrologic record as d13C, however. The utility of d234U was as support for our contention that carbon isotopic variations reflected hydroclimate, which in turn were linked to regional SST.

And why there are large changes in d13C during sometime intervals for which there are smaller changes in SST and in the percent of temperate trees?

We note in the manuscript that pollen obtained from Iberian margin cores is regionally sourced. One cannot compare at fine scales the changes in vegetation occurring over a cave to those occurring regionally. With that said, we do point out and attempt to explain intervals when SST and d13C appear to be decoupled.

---

## Author Comment (AC2) · 25 Jan 2018

As is described below, we are currently in the process of obtaining additional information from the caves and the stalagmites described in this study. We are requesting from the editor additional time in order to fully address the comments made by each of the reviewers. Thus, our responses should be considered preliminary and incomplete versions of the detailed and developed discussion that will accompany a later revised draft of this manuscript.

1.- Chronology. The six speleothems used for this study are complicated samples in terms of growth axis (very variable along the samples), evidences of dissolution, minor

and major hiatus, etc.). In fact, the growth of the six speleothems is very discontinuous and thus making difficult the detection of all the hiatus by U-Th dates. I see two possible ways of improving the chronologies that should be carried out by the authors. First, more dates are necessary in some stalagmites and, this time, analysing a higher amount of calcite would be desirable (I already pointed out this in my previous review. . . 50-150 mg for U-Th dating is insufficient with samples where U concentration is low as it happens here). Sampling a higher amount is possible and necessary to get more accurate dates. Errors of above 2000 years are common in Table 1 and I think they can be improved.

We are acquiring additional dates for select intervals using the larger sample sizes suggested here. These dates will be available in the next few weeks, at which time we will recalculate age models wherever appropriate.

Second, I suggest including some petrographic analyses (thin slides) to help on the identification of hiatus. I am not sure if the authors have done that study since it is not shown but comments on the textures and fabrics are made on 146-157 lines. A figure on this issue in the Supplementary material would be desirable.

We have relied primarily on the U-Th dates to construct the age model. Identifying hiatuses using thin section petrography would not necessarily allow a more accurate age model as the duration of the hiatus cannot be determined independently from the U-Th dates. We therefore feel that the in-depth petrographic analysis of these stalagmites is beyond the scope of this study.

Additionally, I would like to see a figure with all the age models together (an example is provided in Fohlmeister et al., 2012) to show the intervals that are really replicated.

We included age models for individual stalagmites, however this figure will be added to the Supplemental Material in the revised version.

The authors emphasized along the manuscript the good replication of this dataset and

I cannot agree with that. They refer to Fig S2 many times to show replication in d13C records. . ... And in that figure it is evident that replication is really minimal (very short periods and not well reproduced patterns). The authors have to focus their interpretations where chronology was better assessed and replicated and be very cautious where the presence of hiatus was not so well replicated. In fact, sentences like "deposition of multiple stalagmites was punctuated by hiatuses of similar time spans. . ." (lines 181-184) should be avoided (they are not true) and need to be more concrete: I just see one interval where two stalagmites stop growing at the same time, at ca. 100 ka BP.

We did not intend to suggest that the starting and stopping points of each cessation in growth in one stalagmite coincided exactly with hiatuses identified in other stalagmites. Cave hydrology is sufficiently complex so as to make such an event unlikely. Instead, our intention was to note a broad overlap, often during periods of cold and/or dry climates. In the revised version, we will flesh out in detail the intervals of the hiatuses and their relationships to each other and to those in other regional stalagmite records.

Other example: "The reproducibility of carbon isotope ratios between coeval BG stalagmites argues that their d13C values may be viewed as an integrated time series not substantially impacted by inter-sample isotopic offsets" (lines 223-225).

We agree that the amount of overlap is not substantial, but were it occurs, similar d13C values are observed. We will more clearly show this with new figures in the manuscript.

There is also highlighted the coincidence with hiatus in S France and N Spain stalagmites (lines 291-296) and this is not always true (Figure 6).

As mentioned above, we will expand our discussion of the timing of hiatuses in the Portuguese and other regional records. We will note both the intervals when hiatuses do not agree, and also emphasize what we consider to be substantial similarities in the overall timing of hiatuses, particularly given the individual hydrologic characteristics and altitudes? of each cave, differences in regional climate, and the distance between

the cave sites.

2.- Present-day cave environment. To my knowledge, this is the first time paleoclimate records from these two caves are presented. Then, it should be mandatory to understand present-day processes that would help to interpret past records. For example, distinguishing if the correlation of hiatus is with cold or with dry events (or both) is important and must be supported by more present-day data.

As mentioned in our response to a comment by Reviewer 1, we are adding to the Supplemental Material maps of both caves. BG has never been mapped but we are currently working with an experienced team of cavers to create a map suitable for publication. A previously constructed map of GCL will be adapted for publication. More detail on the ages and isotopic compositions of the bedrock hosting each cave will be added, as will isotopic compositions of vegetation and soil organic carbon. In addition, we have recently (January 2018) visited both caves and collected additional environmental monitoring data (temperature, barometric pressure, humidity, and drip rates) acquired via data loggers over the past 1-3 years, and dripwater obtained over the recent months. These new cave monitoring data will be included in the revised manuscript. There have been two years of below average rainfall and drip rates have decreased accordingly during this interval. However, we have no means of examining dramatic reductions in ocean and atmospheric temperature, such as marked Heinrich stadials, would have impacted rainfall or infiltration at these cave sites. We further argue that – while necessary and helpful – modern conditions might not be of great help when investigating glacial-interglacial dynamics, because the last 100-150 years have been dramatically affected by anthropogenic factors ($CO_2$ increase, aerosol emissions preferably in the NH, etc., e.g. Ridley et al. 2015, Nat Geosc.) so as to render extrapolations of modern correlations of above-cave and in-cave changes hardly comparable on such long perspectives.

The authors need to understand what is happening today regarding calcite precipitation in the cave. Does it happens the whole year or focused in the rainy season? Is it more

abundant during warmer years? In lines 238-239 it is said that "any seasonal biases in calcite crystallization remain poorly constrained". Then, how can they link the data to NAO that is a winter process? I think that some interpretations will be better supported by more monitoring data.

Despite continued monitoring, many questions regarding seasonality of calcite deposition remain poorly constrained. We have just obtained plate-grown calcite from BG and are currently measuring its isotopic composition; these data will be included in the revision. We also agree with this reviewer (and Reviewer 4) that linking the Portuguese stalagmite record to the NAO is unnecessary given the scope of this study and we are minimizing this portion of the discussion in the manuscript. We note however, that the NAO influence can be of importance even in summertime (Ogi et al. 2003, GRL; Linderholm et al. 2009, J. Quat. Sci.) and that the question about NAO impact at BG needs careful assessment.

Besides, the switch of 2 per mil in the d13C record from sample CGL6 "for ease of comparison to SST" needs a justification in the text, not a simple note in the figure caption. I do not think such shift in measured values is justified at all without a deeper understanding of the cave environment (soil, host rock, etc).

It is conceivable that the one stalagmite from GCL was impacted by different vegetation over the cave or from secondary effects impacting dripwater. Carbon isotopic values can alter dramatically (several permil) if kinetics are involved (Mühlinghaus et al. 2007) and it is conceivable that drying could lead to a more open system, increased PCP and lower drip rates – all forcing higher d13C values. It is impossible to know what the soil or vegetation over either cave was like at 200 ka. However, the bedrock d13C values of both caves have been measured and this alone cannot account for the full offset with BG. We present the original (unshifted) values of this stalagmite on the same plot and only shift the d13C values in order to better match the SST trend. We therefore feel this is an appropriate means of presenting these data, but will include a more thorough discussion and justification in the revision.

3. Representation of data. Finally, I also have some concerns on the representation of data versus age. In general, I find too "optimistic" sentences or interpretations in the text that are not always easy to seen in the figures. A good example is Figure 6 that is used to emphasize the excellent correspondence of d13C from the stalagmites with other records but the scale does not permit to see it!! Examples: how can we see the positive change during the YD (lines 299-300)? How can we see the hiatus at 80-78 ka (lines 292-293)? What about the "effective moisture from 170-160 ka and 145-135 ka? (lines 303-304). Figure 6 needs more ticks in the x-axis to follow the text and some dashed lines or bars to help the reader to find in the figure the events indicated in the text.

This is good point and Figure 6 will be modified as suggested. In addition, a new figure will be added to the manuscript that zooms in on these intervals to show the overlap.

Regarding representation of data, I also missed some other records that are cited and compared in the text several times, such as Villars cave or many other marine records. Fig. 8 where a zoom is shown for two different intervals would be the place to include those other records. If not, the reader has to go to previous references to compare visually other figures with this new dataset. For the YD, for example, there are many other records available.

We will modify this figure as suggested, and will also expand the scope of Figure 1 to include Villars Cave.

Additionally, I have not found in the text any explanation about the representation of pollen data. Is that a combination of records? A stack? How is it made? And regarding the representation of ice cores, why do not use the "real" ice core for the beginning of the record? The older part can be compared to the synthetic curve, but for the 0-125 ka I suggest to include NGRIP record.

The pollen data are indeed from multiple cores (as listed in Figure 6), but this point will be made more clear in the revision. Per the reviewer's suggestion, original NGRIP data

will be used from 0-125 ka, while the synthetic Greenland record will be used for the remainder.

Minor remarks: - line 285 and line 293. Why Fig. 2?? This is certainly a mistake, I am afraid.

This change will be made.

- line 119-120: explain the correction you did using cave drip water

Approximately 2L of dripwater was collected over several months at BG from the drip that fed the stalagmites analyzed in this study. This water was analyzed for its 230Th/232Th ratio at the University of New Mexico Radiogenic Isotope Laboratory. A more detailed description of these methods will be included in the revision.

- Table S1.There are many reversals not explained in the text.

Including the error windows, there are two dates are stratigraphically reversed and these did not impact the age model. Nonetheless, additional dates are currently being obtained and a more detailed discussion of the U-Th dates will be included in the revised version of the text.

---

## Author Comment (AC3) · 25 Jan 2018

Denniston et al provide a new and long $\delta13C$ and d234u reconstruction of hydroclimate from two caves in Portugal. The stalagmites are securely dated, but have many hiatuses which may be related to climatic variation. In general, warm conditions are associated with lower $\delta13C$ values, suggesting enhanced soil productivity and or decreased prior calcite precipitation, among several hypotheses for the controls on $\delta13C$. Where available, the d234Ui values show similar variations to the $\delta13C$, suggesting the record is one of effective moisture. I find the paper to be clearly written and well-documented, with a copious degree of reconciliation with the literature that attempts

to integrate multiple lines of evidence for hydroclimatic change over Iberia. The presentation does not require improvements in general. However, I wonder if this paper would have more of an impact if it were 2/3rds the length and focused primarily on the record at hand, and its $\delta$13C correlation to the marine SST records? The level of detail is appreciated but may detract from what is by all other accounts is a great record.

We thank the reviewer for these comments. Given the comments by reviewers 1,2, and 4, however, we have opted to expand the manuscript in order to better develop the data and interpretations.

---

## Author Comment (AC4) · 25 Jan 2018

As is described below, we are currently in the process of obtaining additional information from the caves and the stalagmites described in this study. We are requesting from the editor additional time in order to fully address the comments made by each of the reviewers. Thus, our responses should be considered preliminary and incomplete versions of the detailed and developed discussion that will accompany a later revised draft of this manuscript.

1. Presentation of d18O values: My major concern is that the d18O values are only shown in the supplement although they - as is also acknowledged by the authors them-

selves - show some similarity with speleothem d13C and SST off the Iberian margin. I agree that the interpretation of speleothem d18O values on these long time-scales may be not straightforward, but the same is true for d13C. Actually, speleothem d18O values should even be less influenced by local or drip site-specific effects than d13C values and show a more consistent signal. Thus, I am convinced that a combined presentation and discussion of the d13C and the d18O values (similarities/differences) would be much more informative for the reader and also result in a more robust climate record.

This fair point was raised by other reviewers. d13C is a local signal while d18O is a more (pan)regional one. Analysis of regional precipitation (i.e., the GNIP stations at Lisbon and Porto) reveals a complex paleoclimate signal in precipitation d18O. The pronounced seasonality of rainfall leads to strong statistical relationships between both air temperature and rainfall amount that are difficult to disentangle, particularly when spanning intervals of time, such as are contained within this record, in which the seasonality of precipitation likely changed. Nonetheless, we will add a more developed discussion of the d18O data to the manuscript.

2. Discussion of d13C values: The discussion of the d13C data in terms of climate variability is by far too short. The authors mention several processes potentially affecting speleothem d13C values, but then do not discuss which of these processes they consider most important for the observed orbital and millennial scale variability of their record. Is it vegetation density and, resulting from that, soil pCO2? Is it the degree of PCP? Is it drip rate and the resulting changes in disequilibrium isotope fractionation? Or a combination of all these processes? As mentioned above, a detailed comparison (maybe for individual stalagmites) with the d18O records could provide additional information. Even if it is not possible to identify one or two dominant processes, the discussion must be extended in order to present this important information to the reader.

We will expand the relevant discussion based largely on observations made during

our multi-year cave monitoring program. However, we reiterate that without a considerably expanded (trace elements, dye tracing, daily/weekly drip water sampling) protocol, these questions are difficult to constrain.

3. General presentation of the data: The authors state several times in the MS that the d13C (and d18O) signals recorded in the different stalagmites agree in phases of concurrent growth. I agree that this is an important criterion to test whether the stable isotope signals reflects changes in past climate or are dominated by changes in the local karst system/cave. However, in the current form (Figures 6 and S2), it is almost impossible for the reader to judge how good this agreement really is. Please present the corresponding sections on a different time-scale (in several plots) or maybe even calculate correlation coefficients. This is the only way to clearly present the agreement and the differences in timing, evolution and absolute values.

This point was made by other reviewers, and new figures will be added to the revision to address this shortcoming in the manuscript.

4. Dating: The total record is based on 76 ages, and I absolutely acknowledge the associated amount of work and costs. However, whereas for some stalagmites (BG6LR), a large number of ages were determined, for others (BG68, BG67, BG41) only a few samples were dated (and even "dummy" ages inserted). As some of the records clearly show hiatuses that are not accounted for by the current age models (BG67, BG68, Fig. 5), I strongly suggest to date a few more samples (10 may be enough) to improve the age models and, in particular, to better constrain the timing of the hiatuses.

This point was made by other reviewers, and we are currently obtaining additional dates.

Lines 85ff.: The results of the cave monitoring, in particular the detailed drip rate data should be presented in the results section.

We are creating a new sub-section within Results that is dedicated to our multi-year

cave monitoring data. And as previously noted above, we will include cave maps.

Lines 120-121: Please provide more details about the "isotopic analysis of cave drip water" (methods, results, number of samples, etc.).

This information will be included in the cave monitoring section discussed above.

Line 171: MIS 9 should be MIS 7?

The reviewer is correct and this change has been made.

Line 178 ff.: "The similar carbon isotopic trends and values across many of the areas of overlap argue for a consistent climate signal as the primary driver of isotopic variability (Fig. S2)." As stated in my general comment, the similarity is not visible in the current diagrams. Please provide more detailed plots for the corresponding time intervals.

This point was made by other reviewers, and we will create new plots that better display intervals of overlap.

Line 189ff.: "rate of CO2-degassing from water entering the cave" This is a very common mistake in the speleothem literature. The rate of degassing is always very fast. The degree of degassing, however, which is determined by cave pCO2, has an influence on supersaturation and precipitation rate, which in turn may result in disequilibrium effects. It should also be noted that the degree of disequilibrium will be modulated by drip interval, with long intervals resulting in a higher degree of disequilibrium.

These are fair points and are in agreement with the basic thesis of the study, which is that through any of a number of drivers, changes in stalagmite d13C reflects hydroclimate that is, in turn, linked to SST. We will expand/correct the discussion to reflect Reviewer 4's specific comments here.

Line 199ff.: "Thus, increases in carbon isotopic ratios are interpreted here as primarily reflecting a combination of desaturation of voids above the cave and decreased organic CO2 production within the soil zone, both of which are consistent with a vegetative response to cooler, more arid climates (Baker et al., 1997; Genty et al., 2003)." Although I generally agree with the interpretation of the authors that more positive d13C values probably reflect drier conditions, this conclusion requires more discussion of other potential processes (see my general comment). The only process that is reasonably excluded are changes in vegetation type (C3/C4) based on pollen evidence. All other processes (changes in drip rate, supersaturation, ageing of organic material in the soil, etc.) are mentioned, but not discussed at all.

Our discussion of controls on origins of stalagmite carbon isotopic variability will be expanded considerably.

Line 207ff.: "Decreases in effective precipitation and/or bedrock dissolution rate, both of which are associated with increased aridity, have been tied to elevated speleothem $\delta$234U values (Hellstrom and McCulloch, 2000; Plagnes et al., 2002; Polyak et al., 2012), and are interpreted similarly here." Even if this has been discussed elsewhere, for the interested reader, it would be good to briefly (2-3 sentences) mention the underlying process here.

This portion of the discussion will be expanded in order to better clarify the links between d234U and hydroclimate.

Line 211ff.: "As differences in $\delta$234U values between stalagmites may arise from distinct infiltration pathways, we restrict this part of the analysis solely to stalagmite BG6LR, which represents the longest individual stalagmite record of this time series." The same argument holds true for d13C values, which may also strongly depend on differences in infiltration pathways. Differences between the individual stalagmites (in agreement between d13C and d234U) may even provide additional information about the processes occurring in the karst. Please show all the d234U records.

We attempted to construct a composite figure showing all of the d234U data, but the image was confused by the offsets between individual stalagmites. The only stalagmite for which the comparison of d234U and d13C made sense was BGLR6, which spans

the longest interval of time. We will, however, add the requested figure to the revision.

Line 215ff.: Even if the interpretation of the d18O values may be difficult, I am convinced that they contain important information, which should be presented to the reader and discussed in detail (see general comment).

This point was made by other reviewers, and as discussed above, we will considerably expand this section to address these concerns.

Line 231ff.: "The consistency and coherence among carbon (and oxygen) isotope values of coeval stalagmites . . ." Again, this must be presented in a more comprehensive and quantitative way. In the current form, the reader simply has to believe this statement.

This point was made by other reviewers, and as discussed above, we will considerably expand this section to address these concerns, including through a new figure that focuses on the areas of overlap.

Line 233ff.: ". . . the most notable of which is the shift toward higher $\delta 234$ C values at the MIS 5e/6 transition (âĹij130 ka) in stalagmite BG611 that contrasts with the sharp decrease in carbon isotopic ratios in BG67 (Fig. 6)." The corresponding growth phase in stalagmite BG611 appears very short to me (just a few stable isotope data points). Thus, I would not give too much weight to this section of the record. This again highlights the necessity to present the data on a different age scale better showing the details of the record.

This is a fair point and the wording will be changed to reflect the limited number of data points in BG611.

Line 264: HS11 should be HS6?

The reviewer is correct and this change will be made.

Line 330ff.: "This early interglacial peak . . ." Please highlight the corresponding feature

in Fig. 7. It is not clear to me which peak is meant.

This section of the plot will be highlighted.

Line 333: Fig. 2 should be Fig. 1?

This change will be made.

Line 335ff.: "Next, stalagmite $\delta$13C values are lower during GI 20-22 (MIS 5a/4; 84-72 ka) than in either the Holocene or MIS 5e, suggesting that maximum warmth and precipitation were not coincident with peak summer insolation (âĹij127 ka) (Fig. 6)." I strongly disagree with this statement. In particular on these long time-scales, differences in absolute d13C values should not be interpreted in terms of the warmest/coldest or the driest/wettest period. As the authors acknowledge themselves, a variety of parameters may change on these time-scales (karst properties, vegetation type and density, cave ventilation, etc.). Thus, the absolute values should be interpreted with caution.

The reviewer is right in that we should not over-interpret the records. Even in a data set where consistency is, to some degree, tested by overlapping stalagmites, it could be easily true that there are differences in the absolute values between the stalagmite isotopic ratios over time. We will change the discussion accordingly.

Line 348: ". . . than expected based on the observed scaling with SST (Fig. 7)." This is an interesting point, which should be extended in a revised version of the MS. How good is this scaling for the whole record? It may be interesting to see a scatter plot of speleothem d13C vs. SST. In this context, how about the relation between MIS 7 and MIS 5? In the speleothem record, MIS 7 exhibits lower d13C values than MIS 5, which is not the case in all other climate records presented in the paper (Figs. 6 and 7).

This is an excellent suggestion! We will create a scatter plot of SST and binned d13C values for the revised manuscript.

Line 351ff.: "Alternatively, changes in the nature of the NAO . . ." I would remove the

whole discussion on the NAO, which appears rather speculative to me. The NAO is an inter-annual phenomenon, and even if some studies have suggested persistent phases of NAO+ and NAO- in the past, a discussion on the millennial or even orbital time-scale is difficult. Furthermore, this would provide more space for a detailed presentation and discussion of the d18O values and the potential processes influencing the stable isotope signals.

As previously discussed in our response, we agree with this criticism and will minimize this portion of the discussion in the manuscript. However, as discussed in the manuscript, the potential for changes in NAO mean state has been explored for the last millennium and for stadial/interstadial intervals, so we feel NAO belongs as a part of the discussion of the data.

Line 381ff.: "Differences between the structure of the stalagmite and SST records during some time intervals suggest that land-sea connections across Iberia may have varied temporally and spatially." This statement goes too far (see above).

Discussion of the NAO will be minimized in the manuscript. However, it is clear that climatic boundaries (in the sense of Köppen) are non-stationary and that we have to anticipate variability with respect to maritime or continental influences at a given location or, for Portugal, between Csa and Csb climates.

Line 667ff.: "Conservative errors were added to account for the unknown "true" age of the stalagmite at these points." What do you mean by "conservative" errors? Please explain and motivate in detail how those were defined.

The term "conservative error" describes here an error that is as large as possible without causing stratigraphic inversion with respect to the bounding ages. In other words, the outside edge of the error window in the dummy date allows for a minimum of zero temporal offset (i.e., no hiatus), and this error is assigned to both the + and − side of the dummy age. Although one could consider estimating the dummy age error with the average error of the other dates, we believe that this could suggest higher precision

than is warranted (especially as radiometric errors can vary drastically even on small distance changes). It would be prudent to assume smaller errors that are not based on a measurement. This portion of the age modeling technique will be added to the revision.

Fig. 3: Due to the long residence time of the water in the aquifer above the cave, the d18O signal of precipitation is smoothed (at least to some extent). Thus, instead of monthly means, it would be better to show the inter-annual variability and relationships.

Our multi-year cave monitoring program suggests that the residence time of water above the cave is short, likely days to weeks. This information will be included in the revised and expanded discussion of cave monitoring.

Fig. 4: Rather than showing just one year for NAO+ and NAO-, it would be better to show a mean state of all NAO+ and NAO- years within a specific period (e.g., the last 50 years).

Discussion of the NAO will be minimized in the manuscript, but we will add a third panel to this figure that shows the mean state of NAO over the last 50 years.

Fig. 5: Check labelling of the plots. Some speleothem names are different than in the text.

Thanks for catching this. Will correct the labels.

In addition, it appears to me that some of the samples contain apparent hiatuses (e.g., BG67), which are not resolved by the current dating and, thus, not accounted for by the age models. Therefore, I strongly recommend to determine a few more ages to improve the chronologies of these samples and to better constrain the hiatuses.

As previously discussed, we are obtaining additional dates, however several of these stalagmites are characterized by several changes in drip position that may or may not coincide with a cessation of growth. Short hiatuses (perhaps lasting as little as a few decades) may not be resolved by dating and thus must either be assigned a somewhat

arbitrary duration (as constrained by the age model) or plotted as though growth was continuous.

Suggested additional references discussing climate variability on the Iberian Peninsula and in the Mediterranean as well as the timing of orbital and millennial scale climate change:

We thank the reviewer for these additional references and will incorporate them where appropriate.

---

## Author Response (AR1)

Dear Dr. Combourieu-Nebout,

We submit for your consideration the revised version of our manuscript "A Stalagmite Test of North Atlantic SST and Iberian Hydroclimate Linkages over the Last Two Glacial Cycles". We received from four reviewers detailed and expansive suggestions for improving this study, and we have implemented the vast majority. As a result, this manuscript has been greatly improved and expanded and is now ready for reconsideration by *Climate of the Past.*

In the following section, we address, point-by-point, the comments made by each reviewer. Comments are italicized and our responses immediately follow. Because changes to the manuscript are so extensive and, in many cases, our responses to individual reviewer's suggestions are dispersed throughout the manuscript, we do not describe here in detail the specifics of our responses to every comment, but ask you and the reviewers to evaluate the revision in its entirety.

Sincerely,
Rhawn Denniston

**Reviewer 1**
*There is no well-defined structure to the manuscript.*
We have added additional headings and subheadings, and have substantially expanded the content of the original sections.

*There are too many hypothesis and ideas but with no clear background to support them. There are no descriptions of the caves from where the speleothems were sampled.*
One of the caves described in this study (BG) was unmapped when we originally submitted our manuscript, but we have now obtained a detailed cave map for this publication. Maps and vertical profiles for both caves are now included as a figure in this revision. In addition, considerably more detail on the geology and environmental setting of each cave has now been added.

*The correction factor for one cave is the crustal value and for the other is a value determined from the cave drip water, and the difference is substantial. What is the justification to use different correction factor? What can be the reasons, different host rock, soil type, vegetation? Or maybe determining the correction on present-day drips may not be the correct methodology?*
The GCL samples are not particularly sensitive to the initial Th correction, as we now demonstrate in the manuscript. Whether an initial $^{230}$Th/$^{232}$Th ratio of 4.4 ppm (the default value in many studies) or 13.5 ppm (the value calculated from cave dripwater) is used, the ages are similar, and the resulting ages do not impact our interpretations.

*The authors need to put the Figure of the studied speleothems in the text, not in the supplementary material, and indicate the measured ages on the figure, and where the hiatus are. It is important to add petrographic images ̋showing the altered region and regions of hiatus.*
Images of the stalagmites has been moved to the body of the text and includes the U/Th dates and demarcations of the zones of alteration.

*The d18O record follows closely the d13C record. The similar pattern suggests that d18O is also reflecting temperature and humidity, or storm track changes. The authors need to elaborate on*

*this, not to conclude that many factors influence d18O and they include a sentence saying that d18O may be influenced by kinetic effects and evaporation. . ... If evaporation and kinetics would be a major process why there is a good correlation with d13C. These kinds of sentences need to be properly discussed. Thus although it is correct that many factors influence d18O, it is also true for d13C.*

We have substantially expanded the discussion of the drivers of d$^{13}$C and d$^{18}$O values and variability.

*The authors measure the isotopic composition of precipitation and cave water, but prefer not to discuss the d18O of the speleothems, this is strange.*

We have dramatically expanded our discussion of oxygen isotopic ratios in the stalagmites as well as in precipitation and cave dripwater. This includes a discussion of isotopic values of dripwater, plate-grown calcite, bedrock, and vegetation, with the associated data in a supplemental table.

*Why d234 is only shown for part of the record in Figure 6. I would like to see on Fig 6, superimposed also the d18O record.*

Offsets between d$^{234}$U values in individual stalagmites complicate their integration into a single cohesive time series. The only stalagmite for which the comparison of d$^{234}$U and d$^{13}$C made sense was BGLR6, which spans the longest interval of time, and this figure is now included in the text. We have, however, plotted each stalagmite's d$^{234}$U vs its d$^{18}$O and d$^{13}$C and placed this in the Supplemental Material.

*It is clear that during the termination MIS6 to MIS 5 and a more coherent discussion is needed, not just hypothesis and suddenly bring d18O to explain seasonal biases.*

We have expanded the discussion of the 6/5e transition and also go into greater depth regarding the origin of oxygen isotopic values and variability.

*Did the authors performed Hendy test on those speleothems, do verify which of them might have not form in isotopic equilibrium since the repetition test does not work?*

While we are not convinced that the Hendy Test is a reliable means of assessing equilibrium crystallization (as demonstrated in Dorale and Liu, 2009), we performed Hendy tests to address this comment. The results of these analyses are presented in detail in the revision.

*The manuscript is rather confused and a Table showing periods of non-growth can help. Did the authors take into consideration the error on the ages and age model in the final correlations with other proxies in Figs. 6 and 7?*

The data were presented based solely on the age models. We did not add another table as addressing the reviewers' concerns required increasing the number of figures. However, we have attempted to more clearly represent the timing of hiatuses in the text.

*The authors don't explore the very good and interesting data. The discussion is missing explanation on the correlation between d13C and 234U.*

We now provide a more thorough explanation of the links between d$^{234}$U and d$^{13}$C/d$^{18}$O in the revised discussion. As we point out, the d$^{234}$U data are not meant to represent the same sort of

fine-scale paleohydrologic record as d$^{13}$C, but serve instead to support our contention that carbon and oxygen isotopic variations reflect hydroclimate, which in turn are linked to regional SST.

*And why there are large changes in d13C during sometime intervals for which there are smaller changes in SST and in the percent of temperate trees?*
This is an interesting question that we address in the manuscript. We have tried to raise this and related questions more clearly in the revision. The most likely answer is that the pollen is sourced from a large region while the stalagmites record changes at the scale of a single cave.

**Reviewer 2**
*Chronology. The six speleothems used for this study are complicated samples in terms of growth axis (very variable along the samples), evidences of dissolution, minor and major hiatus, etc.). In fact, the growth of the six speleothems is very discontinuous and thus making difficult the detection of all the hiatus by U-Th dates. I see two possible ways of improving the chronologies that should be carried out by the authors. First, more dates are necessary in some stalagmites and, this time, analysing a higher amount of calcite would be desirable (I already pointed out this in my previous review. . . 50-150 mg for U-Th dating is insufficient with samples where U concentration is low as it happens here). Sampling a higher amount is possible and necessary to get more accurate dates. Errors of above 2000 years are common in Table 1 and I think they can be improved.*
We acquired several additional dates using the larger sample sizes suggested here. These dates refined, to some degree, the age models, which were recalculated using these new dates.

*Second, I suggest including some petrographic analyses (thin slides) to help on the identification of hiatus. I am not sure if the authors have done that study since it is not shown but comments on the textures and fabrics are made on 146-157 lines. A figure on this issue in the Supplementary material would be desirable.*
We have relied primarily on the U-Th dates to construct the age model. Identifying hiatuses using thin section petrography would not necessarily allow a more accurate age model as the duration of the hiatus cannot be determined independently from the U-Th dates. We therefore feel that the in-depth petrographic analysis of these stalagmites is beyond the scope of this study. We have, however, included in the supplemental material images of several hiatuses to allow the reader a better sense of their macroscopic petrography.

*Additionally, I would like to see a figure with all the age models together (an example is provided in Fohlmeister et al., 2012) to show the intervals that are really replicated.*
A figure showing depths versus ages (with errors) for each stalagmite has now been added to the Supplemental Material.

*The authors emphasized along the manuscript the good replication of this dataset and I cannot agree with that. They refer to Fig S2 many times to show replication in d13C records. . .. And in that figure it is evident that replication is really minimal (very short periods and not well reproduced patterns). The authors have to focus their interpretations where chronology was better assessed and replicated and be very cautious where the presence of hiatus was not so well replicated. In fact, sentences like "deposition of multiple stalagmites was punctuated by hiatuses of similar time spans. . ." (lines 181-184) should be avoided (they are not true) and need to be*

*more concrete: I just see one interval where two stalagmites stop growing at the same time, at ca. 100 ka BP.*

We have expanded the discussion related to the overlap of coeval stalagmites, including isotopic values and hiatuses. In addition, we added to the supplemental material a figure showing the carbon and oxygen time series for coeval stalagmites.

*Other example: "The reproducibility of carbon isotope ratios between coeval BG stalagmites argues that their d13C values may be viewed as an integrated time series not substantially impacted by inter-sample isotopic offsets" (lines 223-225).*

Please see our response to the previous comment.

*There is also highlighted the coincidence with hiatus in S France and N Spain stalagmites (lines 291-296) and this is not always true (Figure 6).*

As mentioned above, we have expanded our discussion of the timing of hiatuses in the Portuguese and regional records, and hope that this makes more clear the general overlap of hiatuses in some but not all intervals.

*Present-day cave environment. To my knowledge, this is the first time paleoclimate records from these two caves are presented. Then, it should be mandatory to understand present-day processes that would help to interpret past records. For example, distinguishing if the correlation of hiatus is with cold or with dry events (or both) is important and must be supported by more present-day data.*

It is a tall order to require modern cave monitoring to interpret hiatuses from the last glacial. Climatic boundary conditions have changed dramatically since the Pleistocene. However, we have attempted to address this question in the revised manuscript.

*The authors need to understand what is happening today regarding calcite precipitation in the cave. Does it happens the whole year or focused in the rainy season? Is it more abundant during warmer years? In lines 238-239 it is said that "any seasonal biases in calcite crystallization remain poorly constrained". Then, how can they link the data to NAO that is a winter process? I think that some interpretations will be better supported by more monitoring data.*

We now include detailed discussion of the isotopic composition of dripwater and plate-grown calcite from BG. However, the reviewer's point is a good one and we have dramatically reduced the discussion of the NAO in the revision.

*Besides, the switch of 2 per mil in the d13C record from sample CGL6 "for ease of comparison to SST" needs a justification in the text, not a simple note in the figure caption. I do not think such shift in measured values is justified at all without a deeper understanding of the cave environment (soil, host rock, etc).*

With the new U/Th dates on GCL6, the resulting age model has essentially eliminated this offset. We have expanded the discussion of these considerations, however, by including bedrock $d^{13}C$ values and modeled stalagmite $d^{13}C$ values. The vegetation over the cave has been replaced by agricultural eucalyptus in recent years and very little active calcite is forming in GCL. As a result, it is difficult to model stalagmite carbon isotopic values. However, bedrock $d^{13}C$ values between BG and GCL are only ~1‰ apart.

*Representation of data. Finally, I also have some concerns on the representation of data versus age. In general, I find too "optimistic" sentences or interpretations in the text that are not always easy to seen in the figures. A good example is Figure 6 that is used to emphasize the excellent correspondence of d13C from the stalagmites with other records but the scale does not permit to see it!! Examples: how can we see the positive change during the YD (lines 299-300)? How can we see the hiatus at 80-78 ka (lines 292-293)? What about the "effective moisture from 170-160 ka and 145-135 ka? (lines 303-304). Figure 6 needs more ticks in the x-axis to follow the text and some dashed lines or bars to help the reader to find in the figure the events indicated in the text.*

These are all good points. We have added to the revision a figure that divides the BG record into four shorter intervals and plotted the stalagmite d$^{13}$C and d$^{18}$O against Iberian margin SST.

*Regarding representation of data, I also missed some other records that are cited and compared in the text several times, such as Villars cave or many other marine records. Fig. 8 where a zoom is shown for two different intervals would be the place to include those other records. If not, the reader has to go to previous references to compare visually other figures with this new dataset. For the YD, for example, there are many other records available.*

We tried but were not able to include these data on existing figures without overly complicating them. As the manuscript already has a large number of figures, we have left this concern unaddressed.

*Additionally, I have not found in the text any explanation about the representation of pollen data. Is that a combination of records? A stack? How is it made? And regarding the representation of ice cores, why do not use the "real" ice core for the beginning of the record? The older part can be compared to the synthetic curve, but for the 0-125 ka I suggest to include NGRIP record.*

The pollen data are indeed from multiple cores, and this point is now more clearly made. Per the reviewer's suggestion, original NGRIP data are now used from 0-122 ka, while the synthetic Greenland record is used for the remainder.

*Minor remarks: - line 285 and line 293. Why Fig. 2?? This is certainly a mistake, I am afraid.*

This change has been made.

*line 119-120: explain the correction you did using cave drip water*

A more detailed description of these methods is now included in the revision.

*Table S1.There are many reversals not explained in the text.*

Table 1 has now been moved to the body of the manuscript, and the expanded figure of images of the stalagmites that now include U/Th ages demonstrate the stratigraphic consistency of the age model when considering the error envelopes.

Reviewer #3

*Denniston et al provide a new and long δ13C and d234u reconstruction of hydroclimate from two caves in Portugal. The stalagmites are securely dated, but have many hiatuses which may be related to climatic variation. In general, warm conditions are associated with lower δ13C values, suggesting enhanced soil productivity and or decreased prior calcite precipitation, among several hypotheses for the controls on δ13C. Where available, the d234Ui values show*

*similar variations to the δ13C, suggesting the record is one of effective moisture. I find the paper to be clearly written and well-documented, with a copious degree of reconciliation with the literature that attempts to integrate multiple lines of evidence for hydroclimatic change over Iberia. The presentation does not require improvements in general. However, I wonder if this paper would have more of an impact if it were 2/3rds the length and focused primarily on the record at hand, and its δ13C correlation to the marine SST records? The level of detail is appreciated but may detract from what is by all other accounts is a great record.*

We thank the reviewer for these comments. Given the comments by reviewers 1,2, and 4, however, we have opted to expand the manuscript in order to better develop the data and interpretations.

**Reviewer 4**

*Presentation of d18O values: My major concern is that the d18O values are only shown in the supplement although they - as is also acknowledged by the authors themselves - show some similarity with speleothem d13C and SST off the Iberian margin. I agree that the interpretation of speleothem d18O values on these long time-scales may be not straightforward, but the same is true for d13C. Actually, speleothem d18O values should even be less influenced by local or drip site-specific effects than d13C values and show a more consistent signal. Thus, I am convinced that a combined presentation and discussion of the d13C and the d18O values (similarities/differences) would be much more informative for the reader and also result in a more robust climate record.*

As previously discussed, the origins of carbon and oxygen isotopic values and variability are explored in considerably greater detail in the revision.

*Discussion of d13C values: The discussion of the d13C data in terms of climate variability is by far too short. The authors mention several processes potentially affecting speleothem d13C values, but then do not discuss which of these processes they consider most important for the observed orbital and millennial scale variability of their record. Is it vegetation density and, resulting from that, soil pCO2? Is it the degree of PCP? Is it drip rate and the resulting changes in disequilibrium isotope fractionation? Or a combination of all these processes? As mentioned above, a detailed comparison (maybe for individual stalagmites) with the d18O records could provide additional information. Even if it is not possible to identify one or two dominant processes, the discussion must be extended in order to present this important information to the reader.*

As mentioned above, we have substantially expanded the discussion of isotopic values and variability.

*General presentation of the data: The authors state several times in the MS that the d13C (and d18O) signals recorded in the different stalagmites agree in phases of concurrent growth. I agree that this is an important criterion to test whether the stable isotope signals reflects changes in past climate or are dominated by changes in the local karst system/cave. However, in the current form (Figures 6 and S2), it is almost impossible for the reader to judge how good this agreement really is. Please present the corresponding sections on a different time-scale (in several plots) or maybe even calculate correlation coefficients. This is the only way to clearly present the agreement and the differences in timing, evolution and absolute values.*

As mentioned above, this issue has been addressed through the addition of new figures and an expansion of the relevant discussion in the manuscript.

*Dating: The total record is based on 76 ages, and I absolutely acknowledge the associated amount of work and costs. However, whereas for some stalagmites (BG6LR), a large number of ages were determined, for others (BG68, BG67, BG41) only a few samples were dated (and even "dummy" ages inserted). As some of the records clearly show hiatuses that are not accounted for by the current age models (BG67, BG68, Fig. 5), I strongly suggest to date a few more samples (10 may be enough) to improve the age models and, in particular, to better constrain the timing of the hiatuses.*
As mentioned above, additional dates were obtained and all age models were re-evaluated based on these new data.

*Lines 85ff.: The results of the cave monitoring, in particular the detailed drip rate data should be presented in the results section.*
Drip rates are discussed (briefly) and presented in the associated figure.

*Lines 120-121: Please provide more details about the "isotopic analysis of cave drip water" (methods, results, number of samples, etc.).*
These data are presented in considerably more detail in the revision, and a supplemental table with this information is now included.

*Line 171: MIS 9 should be MIS 7?*
The reviewer is correct and this change has been made.

*Line 178 ff.: "The similar carbon isotopic trends and values across many of the areas of overlap argue for a consistent climate signal as the primary driver of isotopic variability (Fig. S2)." As stated in my general comment, the similarity is not visible in the current diagrams. Please provide more detailed plots for the corresponding time intervals.*
This change has been made as mentioned above.

*Line 189ff.: "rate of CO2-degassing from water entering the cave" This is a very common mistake in the speleothem literature. The rate of degassing is always very fast. The degree of degassing, however, which is determined by cave pCO2, has an influence on supersaturation and precipitation rate, which in turn may result in disequilibrium effects. It should also be noted that the degree of disequilibrium will be modulated by drip interval, with long intervals resulting in a higher degree of disequilibrium.*
We have changed this phrase in the manuscript.

*Line 199ff.: "Thus, increases in carbon isotopic ratios are interpreted here as primarily reflecting a combination of desaturation of voids above the cave and decreased organic CO2 production within the soil zone, both of which are consistent with a vegetative response to cooler, more arid climates (Baker et al., 1997; Genty et al., 2003)." Although I generally agree with the interpretation of the authors that more positive d13C values probably reflect drier conditions, this conclusion requires more discussion of other potential processes (see my general comment). The only process that is reasonably excluded are changes in vegetation type (C3/C4)*

*based on pollen evidence. All other processes (changes in drip rate, supersaturation, ageing of organic material in the soil, etc.) are mentioned, but not discussed at all.*
As mentioned above, our discussion of these effects has been substantially expanded.

*Line 207ff.: "Decreases in effective precipitation and/or bedrock dissolution rate, both of which are associated with increased aridity, have been tied to elevated speleothem δ234U values (Hellstrom and McCulloch, 2000; Plagnes et al., 2002; Polyak et al., 2012), and are interpreted similarly here." Even if this has been discussed elsewhere, for the interested reader, it would be good to briefly (2-3 sentences) mention the underlying process here.*
The relevant discussion has been substantially expanded.

*Line 211ff.: "As differences in δ234U values between stalagmites may arise from distinct infiltration pathways, we restrict this part of the analysis solely to stalagmite BG6LR, which represents the longest individual stalagmite record of this time series." The same argument holds true for d13C values, which may also strongly depend on differences in infiltration pathways. Differences between the individual stalagmites (in agreement between d13C and d234U) may even provide additional information about the processes occurring in the karst. Please show all the d234U records.*
A new plot has been added to the supplemental material with d$^{234}$U values plotted against d$^{13}$C for all stalagmites.

*Line 215ff.: Even if the interpretation of the d18O values may be difficult, I am convinced that they contain important information, which should be presented to the reader and discussed in detail (see general comment).*
As mentioned above, the discussion relating to the values and variability of speleothem d$^{18}$O has been substantially expanded.

*Line 231ff.: "The consistency and coherence among carbon (and oxygen) isotope values of coeval stalagmites . . ." Again, this must be presented in a more comprehensive and quantitative way. In the current form, the reader simply has to believe this statement.*
This step has been taken. Please see above.

*Line 233ff.: ". . . the most notable of which is the shift toward higher δ234 C values at the MIS 5e/6 transition ( 130 ka) in stalagmite BG611 that contrasts with the sharp decrease in carbon isotopic ratios in BG67 (Fig. 6)." The corresponding growth phase in stalagmite BG611 appears very short to me (just a few stable isotope data points). Thus, I would not give too much weight to this section of the record. This again highlights the necessity to present the data on a different age scale better showing the details of the record.*
This is a fair point and the wording has been changed to reflect the limited number of data points in BG611 and the associated uncertainties.

*Line 264: HS11 should be HS6?*
The reviewer is correct and this change has been made.

*Line 330ff.: "This early interglacial peak . . ." Please highlight the corresponding feature in Fig. 7. It is not clear to me which peak is meant.*

We feel that this already busy figure would be made even more complicated by denoting the early interglacial peaks with an arrow or asterisk and have thus left this figure unchanged.

*Line 333: Fig. 2 should be Fig. 1?*
This change has been made.

*Line 335ff.: "Next, stalagmite δ13C values are lower during GI 20-22 (MIS 5a/4; 84- 72 ka) than in either the Holocene or MIS 5e, suggesting that maximum warmth and precipitation were not coincident with peak summer insolation ( 127 ka) (Fig. 6)." I strongly disagree with this statement. In particular on these long time-scales, differences in absolute d13C values should not be interpreted in terms of the warmest/ coldest or the driest/wettest period. As the authors acknowledge themselves, a variety of parameters may change on these time-scales (karst properties, vegetation type and density, cave ventilation, etc.). Thus, the absolute values should be interpreted with caution.*
The reviewer is right in that we should not over-interpret the records. Even in a data set where consistency is, to some degree, tested by overlapping stalagmites, it could be easily true that there are differences in the absolute values between the stalagmite isotopic ratios over time. We have changed the discussion accordingly.

*Line 348: ". . . than expected based on the observed scaling with SST (Fig. 7)." This is an interesting point, which should be extended in a revised version of the MS. How good is this scaling for the whole record? It may be interesting to see a scatter plot of speleothem d13C vs. SST. In this context, how about the relation between MIS 7 and MIS 5? In the speleothem record, MIS 7 exhibits lower d13C values than MIS 5, which is not the case in all other climate records presented in the paper (Figs. 6 and 7).*
We created a scatter plot of SST and d$^{13}$C (and d$^{18}$O) values for the revised manuscript. This figure is discussed in the text and presented in the supplemental material.

*Line 351ff.: "Alternatively, changes in the nature of the NAO . . ." I would remove the whole discussion on the NAO, which appears rather speculative to me. The NAO is an inter-annual phenomenon, and even if some studies have suggested persistent phases of NAO+ and NAO- in the past, a discussion on the millennial or even orbital time-scale is difficult. Furthermore, this would provide more space for a detailed presentation and discussion of the d18O values and the potential processes influencing the stable isotope signals.*
As previously discussed in our response, we agree with this criticism and have substantially reduced the discussion of the NAO in the manuscript. However, the potential for changes in NAO mean state has been explored for the last millennium and for stadial/interstadials, and thus we feel that the NAO belongs as a small component of the manuscript.

*Line 381ff.: "Differences between the structure of the stalagmite and SST records during some time intervals suggest that land-sea connections across Iberia may have varied temporally and spatially." This statement goes too far (see above).*
We agree. This statement was removed from the manuscript.

*Line 667ff.: "Conservative errors were added to account for the unknown "true" age of the stalagmite at these points." What do you mean by "conservative" errors? Please explain and motivate in detail how those were defined.*
This information has been added to the manuscript.

*Fig. 3: Due to the long residence time of the water in the aquifer above the cave, the d18O signal of precipitation is smoothed (at least to some extent). Thus, instead of monthly means, it would be better to show the inter-annual variability and relationships.*
While poorly constrained, we argue that the residence time of water above the cave is short, likely weeks. We have thus left the presentation of precipitation isotope data largely the same, albeit with caveats associated with biases introduced by our irregular dripwater sampling schedule.

*Fig. 4: Rather than showing just one year for NAO+ and NAO-, it would be better to show a mean state of all NAO+ and NAO- years within a specific period (e.g., the last 50 years).*
This figure has been changed as suggested.

*Fig. 5: Check labelling of the plots. Some speleothem names are different than in the text.*
We have altered existing plots and created several new ones but will make sure to double check for labeling mistakes.

*In addition, it appears to me that some of the samples contain apparent hiatuses (e.g., BG67), which are not resolved by the current dating and, thus, not accounted for by the age models. Therefore, I strongly recommend to determine a few more ages to improve the chronologies of these samples and to better constrain the hiatuses.*
As mentioned above, additional dates were obtained and growth/age models recalculated. Also the manuscript now includes a more expansive discussion of issues surrounding short-lived hiatuses and their impact on age models.

*Suggested additional references discussing climate variability on the Iberian Peninsula and in the Mediterranean as well as the timing of orbital and millennial scale climate change:*
We thank the reviewer for these additional references and have incorporated them where appropriate.

---

## Referee Report (RR1)

Review of cp-2017-146 "A Stalagmite Test of North Atlantic SST and Iberian Hydroclimate Linkages over the Last Two Glacial Cycles" by Denniston et al..

The authors present a very interesting speleothem record from the west Iberian Peninsula that covers most of the last 250 kyr. The climate in this region is affected by the course and strength of the westerlies during the winter season and is therefore well chosen to conduct this study. The record is unique as it is the first terrestrial record from the west Iberian Peninsula with an independent chronology that covers the last 7 marine isotope stages. This record definitely needs to be published. However, I have a few concerns regarding the presentation of the sample positions for the isotopes and the dating on the stalagmites that prohibit me from assessing the quality of the records. Furthermore, the structure of the manuscript and figures need to be improved, and I have several additional comments that need to be addressed.

1) From figure 6 it can be seen that besides stalagmites BG67, and BG6LR, also GCL6, and BG66 (the part dated at 219 kyr) show evidence of recrystallization at the growth axis. This does not necessarily mean that the record cannot be trusted, because it is unclear when the recrystallization took place (i.e. it could be shortly after initial deposition). Nevertheless, it requires caution with the sampling for the chronology and for the C and O isotopes. Therefore, it is essential that the sample positions of both dating and isotope samples need to be shown clearly. Where sampling was done at the growth axis in recrystallized parts these should be replicated by sampling on the left or right of the growth axis. This is not necessary for the entire stalagmite, but it has to be shown that the isotope signals and chronology are robust and not affected by recrystallization.

2) The structure of the manuscript needs to be improved, I cannot identify a clear red line that is followed through in sections 4 and 5. I believe the paper will be easier to read if the interpretation of the proxies ($\delta^{13}$C, $\delta^{18}$O, $\delta^{234}$U, and growth intervals / hiatus) are set in a "Results + interpretation" section (what is now section 4). In section 5 environmental links can be discussed with other paleoclimate records, and I would suggest to divide this in first order based on timescales and 2$^{nd}$ order the proxies followed by a short intermediate conclusion:

   a. Environmental links on orbital timescales;
      i. Growth intervals / hiatuses
      ii. $\delta^{13}$C + $\delta^{234}$U
      iii. $\delta^{18}$O
      iv. Conclusion
   b. Environmental links during Greenland stadials / Heinrich events;
      i. Growth intervals / hiatuses
      ii. $\delta^{13}$C + $\delta^{234}$U
      iii. $\delta^{18}$O
      iv. Conclusion
   c. Environmental links during DO events;
      i. Growth intervals / hiatuses
      ii. $\delta^{13}$C + $\delta^{234}$U
      iii. $\delta^{18}$O
      iv. Conclusion

3) GNIP data from Porto is used to show relationships between the $\delta^{18}O$ composition of meteoric rainfall and rainfall amount and air temperature. Porto is not indicated on the map in Fig. 1. Importantly it is located 200 km north of the cave sites and experiences a different type of climate with over 1250 mm of annual precipitation, i.e. 750 mm more than at the cave sites. If there are GNIP stations south of the cave sites these are more likely to provide useful information for the interpretation of the $\delta^{18}O$ as the data from Porto (perhaps Lissbon??). The relation between $\delta^{18}O$ and air temperature (i.e. a slope of 0.2‰/°C) cannot be simply extrapolated to the cave site, as these relationships are often site-specific.

Other important comments:

Line 107: The altitudes of the caves are not indicated.

Line 225: I strongly suggest to restructure this to "Results + interpretation"

Line 269 "The second portion of the Hendy Test":
This should be discussed in this paragraph but it is not. Instead the authors continue to describe factors that affect the $\delta^{18}O$ composition of meteoric precipitation, and only come back to this point in lines 309-319. Please restructure and use sub-headers in this section like:
    4.3. Assessing isotope equilibrium
            4.3.1 Hendy tests
            4.3.2. Modeled isotope values
            4.3.3. Replication
    4.4. Interpretation $\delta^{13}C$
    4.5. Interpretation $\delta^{18}O$
    4.6. Interpretation $\delta^{234}U$

Line 369-373:
Is this not in contrast with what is written in section 5 that there are large shifts from arboreal to semi-desert vegetation types? Or does the semi-desert vegetation consist of shrubs and little grasses?

Line 468:
I'm not sure what the authors mean by increasing the age model by 4 and 1.3 kyr? Simply shifting the age depth model by 4 and 1.3 kyr? If the latter, this raises the question whether this is allowed by the age-depth model, because especially 4kyr is really a lot, and based on the uncertainties of the Th/U ages this cannot be done. The Th/U age uncertainties are much smaller for this stalagmite. Also the age-depth model is already an interpretation based on the COPRA algorithm, so some stratigraphic depths associated with a Th/U age may already be interpreted as older or younger as given in Table 1. If the authors seek an objective method to tune the two timeseries I would suggest to use ISCAM (Fohlmeister et al. 2012).

Line 513-514 "while hiatuses….<13.7°C).":
This is not supported by the BG record. There are many low insolation phases with speleothem growth, and high insolation phases that coincide with an hiatus. I find the relation between the occurrence of hiatuses and NH summer insolation for the BG record weak.

Line 521:

The $\delta^{13}$C record is not similar to the NH summer insolation apart from the last 50 kyr, and maybe two more lows around 220 and 150 kyr. I think this can be deleted.

Line 529-532:
"although it.......be involved." Can be deleted, it is speculative and it doesn't lead to any conclusion. It is sufficient to write "The origin of this high variability is unclear. Replication of the Holocene portion of this record currently underway will help address this question (Thatcher et al., 2018).

Line 565-566:
Antarctic $\delta$D and CH4 records are not mentioned anywhere else in the text, which is focused on the climate of the Western Iberian Peninsula, so this is not important for this study and can be deleted.

Line 588:
This is incorrect. There is a NAO reconstruction available from West Greenland that covers the last 5200 years (Olsen et al., 2012), and a Holocene speleothem record form Morocco that covers the time period from 11.5 to 2.6 kyr is interpreted in terms of NAO as well (Wassenburg et al., 2016). These two references should be mentioned here as well.

Figure 1:
Porto is not indicated on the map.

Figure 6:
Scale bars are missing.

Figure 9:
Why not plotting the records with the proxy uncertainty translated in time? This would be very useful in order to assess whether the records replicate or not.

Figure 10:
I strongly suggest to plot the proxy uncertainties here as well to facilitate comparison with other paleoclimate records. In addition, I would suggest to include a graph like in the former Figure 6 that indicates the hiatuses in N Spain and S France with color coding for the specific sites, and please add the hiatuses from BG and GCL records. Right now it is sometimes hard to identify the hiatuses solely based on interruptions of the black line in BG and GCL records.
    Please indicate the timing of YD, HS, and GS in this figure as blue shaded bars like in Fig. 12.

Figure 11:
Please indicate the timing of the GI with shaded bars according to NGRIP. Right now it is rather unclear which peak in the curve is indicated by which number.

Figure 12
Labelling of YD, HS, and GS are missing.

---

## Author Response (AR2)

September 21, 2018

Dear Dr. Combourieu-Nebout,

We submit for your consideration the second revision of our manuscript "A Stalagmite Test of North Atlantic SST and Iberian Hydroclimate Linkages over the Last Two Glacial Cycles". The first revision was evaluated by two reviewers, one of whom argues that the manuscript was acceptable for publication in its current form. The second reviewer offered a series of suggestions for restructuring the manuscript, clarifying some points in the text, and improving some of the figures. We address these edits below.

Sincerely,
Rhawn Denniston

*1) From figure 6 it can be seen that besides stalagmites BG67, and BG6LR, also GCL6, and BG66 (the part dated at 219 kyr) show evidence of recrystallization at the growth axis. This does not necessarily mean that the record cannot be trusted, because it is unclear when the recrystallization took place (i.e. it could be shortly after initial deposition). Nevertheless, it requires caution with the sampling for the chronology and for the C and O isotopes. Therefore, it is essential that the sample positions of both dating and isotope samples need to be shown clearly. Where sampling was done at the growth axis in recrystallized parts these should be replicated by sampling on the left or right of the growth axis. This is not necessary for the entire stalagmite, but it has to be shown that the isotope signals and chronology are robust and not affected by recrystallization.*

We have updated the manuscript text to note the possibility of alteration in parts of BG66 (alteration of GCL6 is already in the manuscript). The discussion of early post-depositional alteration, which does not meaningfully impact U-Th dates, was also added. As the reviewer requested, lines showing sampling traverses have been added to the figure. Isotopic analyses along secondary traverses were performed on two stalagmites. As is evident from the figure added to the Supplemental Material (Fig. S7), the isotopic values are consistent (within analytical errors) between the two transects and the trends are also similar, demonstrating that recrystallization did not materially diminish the paleoclimate information recorded in these sections.

*2) The structure of the manuscript needs to be improved, I cannot identify a clear red line that is followed through in sections 4 and 5. I believe the paper will be easier to read if the interpretation of the proxies ($\delta^{13}C$, $\delta^{18}O$, $\delta^{234}U$, and growth intervals / hiatus) are set in a "Results + interpretation" section (what is now section 4). In section 5 environmental links can be discussed with other paleoclimate records, and I would suggest to divide this in first order based on timescales and $2^{nd}$ order the proxies followed by a short intermediate conclusion:*

*a. Environmental links on orbital timescales; i. Growth intervals / hiatuses  ii. $\delta^{13}C$ + $\delta^{234}U$ iii. $\delta^{18}O$ iv. Conclusion*
*b. Environmental links during Greenland stadials / Heinrich events; i. Growth intervals / hiatuses ii. $\delta^{13}C$ + $\delta^{234}U$ iii. $\delta^{18}O$ iv. Conclusion*
*c. Environmental links during DO events; i. Growth intervals / hiatuses  ii. $\delta^{13}C$ +*

$\delta^{234}$U iii. $\delta^{18}$O iv. Conclusion

Thanks largely to the detailed and thoughtful comments by the four reviewers who evaluated the first version of this manuscript, the structure of this paper evolved substantially into the revision. We appreciate the suggestion by reviewer 2 of the revision (reviewer 5 overall) regarding the format of our presentation, but feel that it is sufficiently clear and concise in its current form.

*3) GNIP data from Porto is used to show relationships between the $\delta^{18}$O composition of meteoric rainfall and rainfall amount and air temperature. Porto is not indicated on the map in Fig. 1. Importantly it is located 200 km north of the cave sites and experiences a different type of climate with over 1250 mm of annual precipitation, i.e. 750 mm more than at the cave sites. If there are GNIP stations south of the cave sites these are more likely to provide useful information for the interpretation of the $\delta^{18}$O as the data from Porto (perhaps Lissbon??). The relation between $\delta^{18}$O and air temperature (i.e. a slope of 0.2‰/°C) cannot be simply extrapolated to the cave site, as these relationships are often site-specific.*
This is a fair point. Unfortunately, the GNIP data from Lisbon are insufficiently detailed or long to be of much use in this case, but we have included (and discussed) GNIP data from Vila Real and Portalegre, and added both of these locations to the map. Interestingly, the slope of the air temperature/precipitation d$^{18}$O relationship is similar at all three sites. Thus our argument regarding the limited impact of temperature on stalagmite d18O compositions appears to hold.

*Other important comments: Line 107: The altitudes of the caves are not indicated. Line 225: I strongly suggest to restructure this to "Results + interpretation"*
The altitudes of the caves are not located in the Cave Settings section but are instead in the Environmental Setting section.

*Line 269 "The second portion of the Hendy Test": This should be discussed in this paragraph but it is not. Instead the authors continue to describe factors that affect the $\delta^{18}$O composition of meteoric precipitation, and only come back to this point in lines 309-319. Please restructure and use sub-headers in this section like:*

*4.3. Assessing isotope equilibrium 4.3.1 Hendy tests*
*4.3.2. Modeled isotope values*
*4.3.3. Replication 4.4. Interpretation $\delta^{13}$C 4.5. Interpretation $\delta^{18}$O*
*4.6. Interpretation $\delta^{234}$U*

Once again we appreciate the reviewer's suggestion for restructuring this section of the manuscript but here we also feel that the current structure provides the most straightforward means of presenting these arguments.

*Line 369-373: Is this not in contrast with what is written in section 5 that there are large shifts from arboreal to semi-desert vegetation types? Or does the semi-desert vegetation consist of shrubs and little grasses?*
There is no conflict. C3 vegetation dominated at all times but large proportional changes

were observed in semi-desert vegetation (overall abundances remained low).

*Line 468: I'm not sure what the authors mean by increasing the age model by 4 and 1.3 kyr? Simply shifting the age depth model by 4 and 1.3 kyr? If the latter, this raises the question whether this is allowed by the age-depth model, because especially 4kyr is really a lot, and based on the uncertainties of the Th/U ages this cannot be done. The Th/U age uncertainties are much smaller for this stalagmite. Also the age-depth model is already an interpretation based on the COPRA algorithm, so some stratigraphic depths associated with a Th/U age may already be interpreted as older or younger as given in Table 1. If the authors seek an objective method to tune the two timeseries I would suggest to use ISCAM (Fohlmeister et al. 2012).*
The age models were shifted consistently by these amounts as a method of tuning the overlapping time series to the SST record. The magnitude of both offsets (4 and 1.3 kyr) is smaller than the age uncertainties in the combined stalagmite and SST age models.

*Line 513-514 "while hiatuses....<13.7°C).": This is not supported by the BG record. There are many low insolation phases with speleothem growth, and high insolation phases that coincide with an hiatus. I find the relation between the occurrence of hiatuses and NH summer insolation for the BG record weak.*
We agree; within the larger context of the data, SST plays a more important role than insolation. The text has been changed to reflect this idea.

*Line 521: The $\delta^{13}C$ record is not similar to the NH summer insolation apart from the last 50 kyr, and maybe two more lows around 220 and 150 kyr. I think this can be deleted.*
We agree and have made this change to the manuscript.

*Line 529-532: "although it.......be involved." Can be deleted, it is speculative and it doesn't lead to any conclusion. It is sufficient to write "The origin of this high variability is unclear. Replication of the Holocene portion of this record currently underway will help address this question (Thatcher et al., 2018).*
We agree and have made this change to the manuscript.

*Line 565-566: Antaractic δD and CH4 records are not mentioned anywhere else in the text, which is focused on the climate of the Western Iberian Peninsula, so this is not important for this study and can be deleted.*
We were conflicted about including the Antarctic methane and deuterium data, as well, but decided to leave them in the manuscript to illustrate that the early interglacial peak is not an artifact of a small number of European data sets but is, instead, recognized as a global phenomenon. As a result, we have left this section unchanged.

*Line 588: This is incorrect. There is a NAO reconstruction available from West Greenland that covers the last 5200 years (Olsen et al., 2012), and a Holocene speleothem record form Morocco that covers the time period from 11.5 to 2.6 kyr is interpreted in terms of NAO as well (Wassenburg et al., 2016). These two references should be mentioned here as well.*
These two studies are now cited in the manuscript.

*Figure 1: Porto is not indicated on the map.*

Porto is present on our version of the map. Perhaps there was an issue with figure translation within the CotP system? As discussed above, we have also added the locations of two other GNIP sites.

*Figure 6: Scale bars are missing.*
We have added scale bars to illustrate the differential sizing of each stalagmite.

*Figure 9: Why not plotting the records with the proxy uncertainty translated in time? This would be very useful in order to assess whether the records replicate or not.*
We see the reviewer's point but hesitate to tune the records to each other given the number of stalagmites and the errors on the ages. We feel that the reader can adequately assess the degree of overlap/covariance within the constraints of the approach used for the figures.

*Figure 10: I strongly suggest to plot the proxy uncertainties here as well to facilitate comparison with other paleoclimate records. In addition, I would suggest to include a graph like in the former Figure 6 that indicates the hiatuses in N Spain and S France with color coding for the specific sites, and please add the hiatuses from BG and GCL records. Right now it is sometimes hard to identify the hiatuses solely based on interruptions of the black line in BG and GCL records.*
We agree that color-coded bars to illustrate site-specific hiatuses would be useful, and we have played around with this idea quite a bit. The trade-off in doing so, of course, is that the already busy figure gets increasingly complicated and difficult to make sense of. At the risk of seemingly overly resistant to the helpful suggestions by this reviewer, we again have left this portion of the manuscript unchanged.

*Please indicate the timing of YD, HS, and GS in this figure as blue shaded bars like in Fig. 12.*
Ditto our previous response. The issue is one of balancing clarity against the amount of information provided. Relative to the first version of this manuscript, we have markedly increased the number of figures in order to allow the presentation of this sort of detail.

*Figure 11: Please indicate the timing of the GI with shaded bars according to NGRIP. Right now it is rather unclear which peak in the curve is indicated by which number.*
We appreciate the reviewer's concern but feel that this figure is already complex and adding blue bars will only make it less readable.

*Figure 12 Labelling of YD, HS, and GS are missing.*
Labels have been added to the figure for YD and HS. Labels for GS are not added as they are not the focus of this figure.

Line 426-429: We also added a short discussion of the anomalously low $\delta^{18}O$ values associated with the GI-1 that was absent from our earlier drafts.

---

## Author Response (AR3)

November 1, 2018
Dear Dr. Combourieu-Nebout,

We submit for your consideration the third revision of our manuscript "A Stalagmite Test of North Atlantic SST and Iberian Hydroclimate Linkages over the Last Two Glacial Cycles". The second revision was evaluated by one reviewer, whom made some additional suggestions, one of which required us to run more stable isotopes. We address these edits below (reviewer's comments in italics).

Sincerely,
Rhawn Denniston

1) *The authors made some effort to show that recrystallization did not affect their isotope curve in a meaningful way and show a comparison with isotope data off-axis in Fig. S7. However, I cite from my first review: "BG66 (the part dated at 219 kyr) show evidence of recrystallization at the growth axis". I'm highlighting 219 ka, because this is the oldest part of the entire record and shows an important shift in both carbon and oxygen isotopes that matches a large shift in Portugese margin SST. Instead of replicating this shift, the authors chose a different part of the stalagmite to obtain a second isotope transect off-axis, where potential recrystallization is much less clear. I'm puzzled about the reasons for this, and emphasize the importance of replicating the record between 222.5 to 210 ka. Because this is where the recrystallization in this stalagmite is most obvious. So this still has to be addressed.*

We have now replicated the portion of BG66 that the reviewer identified. A transect was milled parallel to the original sampling sites but outside the zone of alteration that exists within the stalagmite core. The data from this new transect agree quite well with the original time series. These data are included in the relevant figure in the Supplemental Material.

2) *I acknowledge that the authors have improved the structure of the manuscript before the manuscript was sent out to me. I made several suggestions, that I think would improve it further. However, the authors have not adapted the structure of the manuscript. My opinion on this matter has not changed, so it is up to the editor to decide whether the structure of the manuscript is acceptable or not. In either case, these are my suggestions:*

*a. Line 272: This is the point where the structure of the manuscript becomes less clear to me. The section on "hendy tests" is not brought to a conclusion. Instead, the authors start discussing other controls on δ18O and δ13C. This can be VERY easily improved by starting chapter 4.3. by discussing the controls as in lines 273 to 311. And then discuss potential equilibrium / disequilibrium effects. Finishing with a small paragraph that the function of hendy-tests have been questioned. Stating that the signals visible in these stalagmites can be interpreted as a climate signal.*

Please see our response to the next suggested edit.

*b. Another option would be to move the section on controls on d13C and d18O in section 4.3. to section 4.4. This way you can call section 4.3. "assessing equilibrium in speleothem δ18O and δ13C. I think this is the most elegant way to do it.*

We have made the changes to the structure of the manuscript as suggested in (b).

*3) I cannot see the locations of the GNIP stations on the map, but in the author's reply they mention that they do indicate it, so this might be an issue when uploading the manuscript? The addition of especially Portalegre is important, because it is located in the same climatic zone as the caves. The authors mention the correlations with air temperature at all GNIP stations, but not for precipitation amount. Please add this information for all GNIP stations.*

The relationship between precipitation amount and precipitation d18O has been added to the relevant portion of the manuscript text.

*Line 474: "model ages" I think you mean "age model uncertainties"*

We did, in fact, shift the model ages in order to better mesh with the SST data. Age model uncertainties were not impacted.

---

## Author Response (AR5)

Dear Dr. Combourieu-Nebout,

We submit for your consideration the fourth version of our manuscript "A Stalagmite Test of North Atlantic SST and Iberian Hydroclimate Linkages over the Last Two Glacial Cycles", which now contains all of the changes recommended by you based on the last iteration of this work. Your suggestions were insightful and detailed, and we appreciate them.

Sincerely,
Rhawn Denniston

[revised manuscript text omitted]

---

## Author Response (AR6)

November 27, 2018

Dear Dr. Combourieu-Nebout,

We submit for your consideration the fifth version of our manuscript "A Stalagmite Test of North Atlantic SST and Iberian Hydroclimate Linkages over the Last Two Glacial Cycles". We have made the change from "model age" to "age model", as requested, and have also updated the address of one of the co-authors. Setsen Altan-Ochir is now at École Normale Supérieure in Paris. Per the instructions for authors, Table 1 is now a Word file and the captions have been removed from the figures.

Sincerely,
Rhawn Denniston